# Location bias contributes to functionally selective responses of biased CXCR3 agonists

Dylan Scott Eiger [1], Noelia Boldizsar[2], Christopher Cole Honeycutt [2], Julia Gardner [2], Stephen Kirchner[3,4], Chloe Hicks [2], Issac Choi [5], Uyen Pham[1], Kevin Zheng[6], Anmol Warman[2], Jeffrey S. Smith [6,7,8,9,10], Jennifer Y. Zhang [3,11] & Sudarshan Rajagopal [1,5] ✉

Some G protein-coupled receptor (GPCR) ligands act as "biased agonists" that preferentially activate specific signaling transducers over others. Although GPCRs are primarily found at the plasma membrane, GPCRs can traffic to and signal from many subcellular compartments. Here, we determine that differential subcellular signaling contributes to the biased signaling generated by three endogenous ligands of the GPCR CXC chemokine receptor 3 (CXCR3). The signaling profile of CXCR3 changes as it traffics from the plasma membrane to endosomes in a ligand-specific manner. Endosomal signaling is critical for biased activation of G proteins, β-arrestins, and extracellular-signal-regulated kinase (ERK). In CD8 + T cells, the chemokines promote unique transcriptional responses predicted to regulate inflammatory pathways. In a mouse model of contact hypersensitivity, β-arrestin-biased CXCR3-mediated inflammation is dependent on receptor internalization. Our work demonstrates that differential subcellular signaling is critical to the overall biased response observed at CXCR3, which has important implications for drugs targeting chemokine receptors and other GPCRs.

G Protein-Coupled Receptors (GPCRs) are the largest superfamily of membrane proteins, accounting for about 5% of all genes encoded in the human genome[1], and are the target of approximately 35% of all Food and Drug Administration-approved drugs[2]. GPCR signaling is mediated by effectors including G proteins, GPCR kinases (GRKs), and β-arrestins, which modulate the activity of a variety of signaling pathways, like those mediated by cyclic adenosine monophosphate (cAMP), extracellular-signal-regulated kinase (ERK), and protein kinase A (PKA)[3]. GPCR signaling is implicated in a wide range of normal physiologic processes[4], and the dysregulation of GPCRs is associated with various pathologies[5]. Ligand:receptor interactions at GPCRs can preferentially activate certain signaling pathways over others in a

ligand-, receptor- or cell-dependent manner, a phenomenon referred to as "biased agonism" or "functional selectivity"[6]. There is a desire to develop *biased agonists* that selectively activate some signaling pathways over others to generate beneficial physiologic responses while reducing off-target effects. However, the molecular mechanisms underlying biased signaling remain unclear.

Adding to this complexity has been the realization that GPCRs can signal from subcellular compartments with altered signaling profiles, resulting in 'location bias' as an additional mechanism of signaling specificity[7–11]. GPCRs can undergo receptor-mediated endocytosis and be recycled back to the plasma membrane, targeted to lysosomes for degradation, or trafficked to specific subcellular locations[12]. It was

[1]Department of Biochemistry, Duke University, Durham, NC 27710, USA. [2]Trinity College, Duke University, Durham, NC 27710, USA. [3]Department of Dermatology, Duke University, Durham, NC 27707, USA. [4]Department of Molecular Genetics and Microbiology, Duke University, Durham, NC 27707, USA. [5]Department of Medicine, Duke University, Durham, NC 27710, USA. [6]Harvard Medical School, Boston, MA 02115, USA. [7]Department of Dermatology, Brigham and Women's Hospital, Boston, MA 02115, USA. [8]Department of Dermatology, Beth Israel Deaconess Medical Center, Boston, MA 02215, USA. [9]Dermatology Program, Boston Children's Hospital, Boston, MA 02115, USA. [10]Department of Dermatology, Massachusetts General Hospital, Boston, MA 02114, USA. [11]Department of Pathology, Duke University, Durham, NC 27710, USA. ✉e-mail: sudarshan.rajagopal@duke.edu

previously thought that GPCR internalization abolished signaling by limiting the membrane-accessible GPCR pool or via receptor degradation[13]. However, it was later appreciated that GPCRs can activate G protein- and β-arrestin-mediated signaling pathways from both the plasma membrane and endosomes[14–16], and other subcellular compartments, like the Golgi apparatus[17] and endoplasmic reticulum (ER)[18]. Internalized GPCR signaling is an enticing therapeutic target with potential to broaden our ability to manipulate GPCR-mediated physiological processes and disease states[19–21]. However, it is unclear to what extent subcellular signaling contributes to the overall biased signaling exhibited by GPCRs.

The physiologic significance of location bias is difficult to determine as most biased agonists are synthetic ligands. However, chemokine receptors (CKRs) represent a subfamily of GPCRs consisting of approximately 20 receptors and 50 endogenous ligands that interact to regulate many cellular functions like chemotaxis, angiogenesis, and neuromodulation[22]. CKRs are promiscuous in that some receptors bind multiple ligands, and some ligands bind multiple receptors. For example, CXCR3 is a CKR with three endogenous ligands, CXCL9, CXCL10, and CXCL11, and is expressed primarily on effector T cells[23–25]. CXCR3 signaling, like many other CKRs, is primarily mediated by both Gαi- and β-arrestin-dependent pathways[26]. Previous work has shown that CXCL11 is relatively β-arrestin biased compared to CXCL9 and CXCL10, and each chemokine demonstrates distinct abilities to promote receptor-mediated endocytosis[25–27].

With its biased signaling and internalization and central role in regulating T cell biology, we studied CXCR3 and its endogenous ligands to determine how ligand bias extends beyond the plasma membrane to the endosome, with implications for sustained, differential signaling at specific subcellular compartments. We demonstrate that the CXCR3 ligands activate G proteins and β-arrestins differently at the endosome compared to the plasma membrane. Furthermore, downstream signaling responses, like kinase activation and cellular transcription, are differentially regulated by the endogenous ligands in a manner dependent on receptor internalization. We determine that the chemokines differentially modulate transcriptional pathways related to inflammation in primary CD8 + T cells, and demonstrate that internalization is required to fully potentiate the inflammatory response in a mouse model of contact hypersensitivity. We demonstrate how biased GPCR signaling can change as the receptor traffics to a subcellular compartment with important physiological effects, and also highlight how a significant proportion of GPCR functional selectivity is dependent on sustained signaling following receptor internalization.

## Results

### Chemokines promote different amounts of β-arrestin-dependent CXCR3 internalization

We first determined if the biased chemokines of CXCR3 promoted different amounts of receptor-mediated internalization in HEK293 cells. Using bioluminescence resonance energy transfer (BRET), we monitored a luciferase-tagged CXCR3 as it traffics to endosomes with an FYVE domain-tagged mVenus, or away from the plasma membrane using a Myrpalm-tagged mVenus. Consistent with previous studies, the chemokines promoted different degrees of receptor-mediated endocytosis with CXCL11 being the most efficacious ligand (Fig. 1a)[25,28]. β-arrestins are known to interact with a variety of effector proteins, including those involved in endocytosis[29–33]. To determine the role β-arrestins play in receptor internalization at CXCR3, we studied CXCR3 internalization in β-arrestin 1/2 CRISPR KO cells[34,35]. Internalization was abrogated in the absence of β-arrestin 1 and 2 and reintroduction of β-arrestin 1 and/or β-arrestin 2 rescued CXCR3 internalization following stimulation with CXCL10 and CXCL11, but not CXCL9 (Fig. 1b). Using confocal microscopy, we similarly observed an increase in receptor internalization upon rescue with β-arrestin 1 or β-arrestin 2 following

stimulation with CXCL10 and CXCL11 (Fig. 1c). Together, these data demonstrate that the CXCL10 and CXCL11 promote CXCR3 internalization in a β-arrestin-dependent manner.

### Biased G protein activation depends on receptor location

To determine how CXCR3 activates G proteins at the plasma membrane and endosomes, we used a location-specific BRET biosensor to detect GTP-bound Gαi as a measure of G protein activation[36,37] (Fig. 2a, b). Importantly, the endosomal and membrane BRET biosensors were expressed at similar levels (Supplementary Fig. 1a). At the plasma membrane, CXCL11 promoted the most G protein activation followed by CXCL10 and lastly CXCL9 (Fig. 2c), consistent with previous reports[26,38]. All CXCR3 endogenous ligands promoted G protein activation at the endosome (Fig. 2d). The amount of G protein activation was different than that observed at the plasma membrane; specifically, CXCL10 and CXCL11 had nearly identical G protein activation at the endosome but different amounts at the plasma membrane. Furthermore, CXCL11-induced G protein activation decreased in the endosome compared to the plasma membrane, while those of CXCL9 and CXCL10 did not change (Fig. 2e–g), demonstrating that the impact of receptor location on G protein signaling is ligand-specific.

Gαi family members are myristoylated, which localizes these proteins to the plasma membrane[39]. We then tested if the relative change in endosomal G protein activation could be explained by different amounts of *total* G protein present in the endosomes. To do this, we developed a split nanoluciferase assay to determine the *absolute* amount of Gαi present in endosomes, irrespective of Gαi nucleotide status (Fig. 2h). We found that Gαi rapidly translocated to the endosome following stimulation with the CXCR3 ligands, and the total amount of endosomal G protein mirrored a chemokine's ability to induce receptor internalization (Fig. 2i, j). Therefore, although similar amounts of endosomal G protein activation were observed following treatment with CXCL10 and CXCL11, when considering the absolute amount of endosomal G protein, CXCL11 promoted relatively *less* G protein activation than CXCL10. We further normalized these data by the total amount of receptor present in the endosome and similarly found that CXCL9 and CXCL10 are relatively more efficacious at promoting endosomal G protein activation than CXCL11 (Supplementary Fig. 1b). These data demonstrate ligand and location bias in G protein activation, with different levels of G protein activation at the plasma membrane compared to the endosome depending on the agonist.

### CXCR3-mediated cAMP inhibition is partially dependent on receptor internalization

We next studied the effect of inhibiting endocytosis on the intracellular accumulation of cAMP. While Gαs family members activate adenylyl cyclase (AC) to produce cAMP, Gαi family members inhibit AC. We utilized an exchange protein activated by cAMP (EPAC)-based BRET biosensor for cAMP that is ubiquitously expressed in cells[40] (Fig. 3a). Prior to activation of the endogenous Gαs-coupled β2-adrenergic receptor (β2AR), HEK293 cells were preincubated with the CXCR3 ligands, allowing us to measure Gαi activity (Fig. 3a). To inhibit receptor-mediated internalization, we overexpressed a dominant-negative mutant of the GTPase Dynamin (Dynamin K44A), which is required for release of clathrin-coated vesicles from the plasma membrane[41]. Using confocal microscopy, we confirmed that Dynamin K44A inhibited the translocation of membrane-bound CXCR3-GFP:β-arrestin 2-RFP complexes into endosomes (Supplementary Fig. 2a).

Chemokine inhibition of cAMP production mirrored Gαi nucleotide exchange, where CXCL11 and CXCL10 are significantly more potent and efficacious agonists than CXCL9 (Fig. 3b). Expression of Dynamin K44A reduced inhibition of cAMP production following stimulation with CXCL10 and CXCL11, but not CXCL9, reflecting a biased decrease in Gαi-coupled activity (Fig. 3c). However, this result may be due to the low G protein signal generated by CXCL9, and all

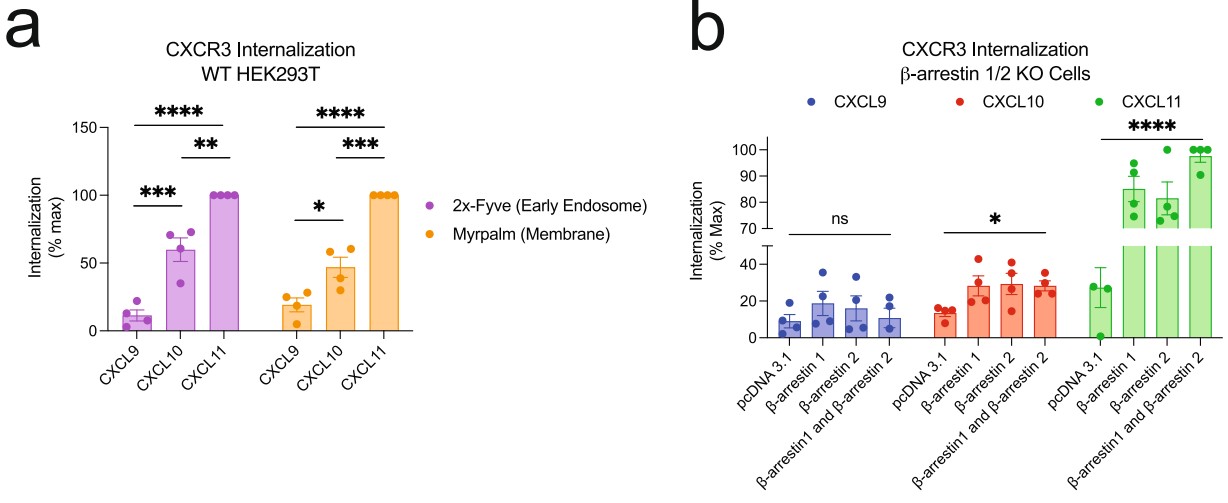

chemokines may demonstrate reduced inhibition of cAMP production at higher and supraphysiologic levels of chemokine. CXCL10 and CXCL11 both demonstrated a ~40% decrease in cAMP inhibition when receptor internalization was inhibited, even though the chemokines were able to promote different amounts of total receptor internalization (Fig. 3d–i). This conclusion is dependent on there being a significant difference in normalized cAMP inhibition by CXCL10 and

CXCL11. Given that this assay directly measures inhibition of cAMP production, which is an amplified response of Gαi activity, detecting real differences between high efficacy agonists is difficult. However, these data demonstrate that both CXCL10 and CXCL11 require receptor internalization to achieve maximal inhibition of cAMP production.

cAMP gradients can exist in micro or nanodomains within the cell, and endosomal cAMP production can be critical for nuclear

**Fig. 1 | CXCR3 receptor-mediated internalization is differentially regulated by biased chemokines and dependent on β-arrestin. a** CXCR3 trafficking to early endosomes using the BRET acceptor 2x-Fyve-mVenus or away from the plasma membrane using Myrpalm-mVenus in HEK293 cells. **b** CXCR3 trafficking away from the plasma membrane using Myrpalm-mVenus in β-arrestin 1/2 knock out cells following transfection of an empty vector (pcDNA 3.1), β-arrestin 1, β-arrestin 2, or both β-arrestin 1 and β-arrestin 2. Data are normalized to maximum signal measured between 25- and 30-min following 100 nM chemokine treatment and are the mean ± SEM, $n = 4$ independent plate-based experiments. ns $P \geq 0.05$,

* $P = 0.01–0.05$, **$P = 0.001–0.01$, ***$P = 0.0001–0.001$, ****$P < 0.0001$ by one-way or two-way ANOVA. For **a**, Tukey's post-hoc testing was conducted between ligands within each BRET acceptor and for **b** Dunnet's post-hoc testing was conducted between pcDNA 3.1 and every other transfection condition within a ligand. **c** Confocal microscopy images of β-arrestin 1/2 knock out cells transfected with CXCR3-mCerulean and either pcDNA 3.1, β-arrestin 1, or β-arrestin 2 following the listed treatment at 100 nM for 45 min. Images are representative of three independent replicates.

translocation of effectors like PKA[42–44]. Using an EPAC BRET biosensor localized to the nucleus, we found that the pattern of cAMP inhibition was nearly identical to that measured globally (Supplementary Fig. 2b–j). These data demonstrate that receptor internalization is critical to the regulation of second messengers across subcellular compartments.

### Chemokines promote differential and location-dependent β-arrestin 2 recruitment and conformation

We next determined if the location-dependent functional selectivity observed in G protein signaling extended to β-arrestins. Consistent with previous studies, CXCL11 induced the most β-arrestin 2 recruitment to the plasma membrane, followed by CXCL10 and CXCL9 (Fig. 4a, b)[26]. While CXCL11 promoted robust and sustained β-arrestin 2 recruitment to endosomes, CXCL10 only weakly and transiently recruited β-arrestin 2, while CXCL9 showed no detectable endosomal recruitment (Fig. 4c, d). GPCR affinity for β-arrestins can be classified as "Class A" GPCRs, which form transient complexes with β-arrestins, while "Class B" GPCRs form tight and long-lived complexes with β-arrestins[45,46]. CXCL9 and CXCL10 promote CXCR3 to behave like a "Class A" GPCR while CXCL11 promotes "Class B" behavior, a phenomenon previously described at other GPCRs[47].

Recent research demonstrated that distinct conformations of β-arrestin mediate specific signaling events like GPCR desensitization, internalization, and effector scaffolding[48–51]. We developed an assay to quantify β-arrestin 2 conformation at specific cellular locations based on a previously described intramolecular fluorescent arsenical hairpin (FlAsH) BRET assay[52]. This modified "complex FlAsH" assay takes advantage of a split nanoluciferase-coupled with FlAsH BRET (Fig. 4e, h), and provides a readout of β-arrestin 2 conformation at specific subcellular locations. We assessed the conformational status of β-arrestin 2 using two previously validated FlAsH constructs, FlAsH 4 and FlAsH 5, which demonstrate preserved β-arrestin recruitment to GPCRs[52]. β-arrestin activation is associated with a ~20° rotation between its N- and C-domains[53]. Given the common location of the BRET acceptor on the β-arrestin 2 C-domain in the FlAsH 4 and FlAsH 5 constructs, these sensors serve as readouts of β-arrestin interdomain twist[54]. We found that the biased ligands of CXCR3 display markedly distinct patterns of FlAsH conformational signatures at both the plasma membrane and the endosome, suggesting that bias in β-arrestin 2 conformation is different at specific subcellular locations (Fig. 4f–g and i–j). While CXCL9 and CXCL10 recruited β-arrestin 2 to the plasma membrane, both chemokines did not induce significant change in β-arrestin 2 conformation at this location. CXCL11-induced distinct FlAsH signatures from CXCL9 and CXCL10 at the plasma membrane (Fig. 4f–g). At the endosome, CXCL10 and CXCL11 induced significant but different changes in β-arrestin 2 conformation while CXCL9 demonstrated no change in conformation, consistent with its inability to recruit β-arrestin 2 to endosomes (Fig. 4i–j). While the β-arrestin 2 conformation demonstrated an increase in BRET signal at the plasma membrane, we observed a decrease in BRET signal at the endosome, which could be due to differences in β-arrestin 2 conformation at the endosome compared to the plasma membrane and/or a change in orientation between β-arrestin 2 and the different location markers. Not only do the chemokines differentially recruit β-arrestin 2

to the plasma membrane and the endosome, but the conformation of β-arrestin 2 is uniquely dependent on both agonist and location, consistent with location bias in β-arrestin activity between agonists.

### Biased signaling profiles of the chemokines changes as the receptor traffics to endosomes

Biased agonism at GPCRs is commonly assessed in terms of the relative activation between G proteins and β-arrestins, and we summarized the above findings using bias plots (Fig. 4k, l)[55,56]. Bias plots allow for simultaneous assessment of relative activity between two assays, and the best fit lines obtained for each chemokine can assess relative bias across the ligands.

Measurements in G protein activation/recruitment and β-arrestin recruitment at different locations are potentially impacted by the biosensor used to detect these events. For example, the absolute change in BRET signal of CXCL11 mediated G protein activation at the plasma membrane and endosome is different, but it is possible that this difference is due to using two different location-specific biosensors, rather than amounts of activated G protein. However, we did not observe such a difference in CXCL10 mediated G protein activation at these two locations. Calculating and comparing the difference of differences (differences between chemokines and locations), removes any potential contribution that the location-specific biosensors may have to detect cellular events (Fig. 4k–l). In the bias plots, we highlight these difference of differences as relative changes in bias, rather than absolute changes in bias. Specifically, the slopes of the bias plots show a different rank order for the three ligands at the plasma membrane versus the endosome. This analysis, which avoids direct comparison of biosensors in different compartments, demonstrates that biosensor location-specific effects cannot account for the observed signaling biases.

At the plasma membrane, we observed that CXCL11 is slightly β-arrestin-biased compared to CXCL10. CXCL9 demonstrated a similar profile to CXCL11, but with partial agonist activity. At the endosome, CXCL11 demonstrated a relative decrease in G protein activation while still effectively coupling to β-arrestin. Conversely, CXCL9 and CXCL10 demonstrated a significant increase in relative G protein activation and simultaneous decrease in coupling to β-arrestin. Together, the relative β-arrestin-biased nature of CXCL11 and the G protein-biased nature of CXCL10 at the plasma membrane was increased in the endosome. CXCL9 acts as a partial, relatively β-arrestin-biased agonist at the plasma membrane, but it becomes significantly more G protein-biased in the endosome (Supplementary Fig. 3). Bias plots do not highlight the absolute amount of signaling across ligands. For example, although CXCL11 is β-arrestin-biased at the endosome, while CXCL9 is G protein-biased at the endosome, CXCL11 activates more absolute amounts of G protein in the endosome than CXCL9. The relative bias between the two ligands is determined when considering both G protein and β-arrestin signaling together. Our analysis provides an assessment of biased signaling which considers the intrinsic efficacy and potency of one ligand to signal across multiple pathways in reference to another ligand.

### CXCR3 signaling from endosomes differentially contributes to cytoplasmic and nuclear ERK activation

We next investigated the activation of the mitogen-activated protein kinase (MAPK) pathway through ERK 1/2 phosphorylation (pERK), a

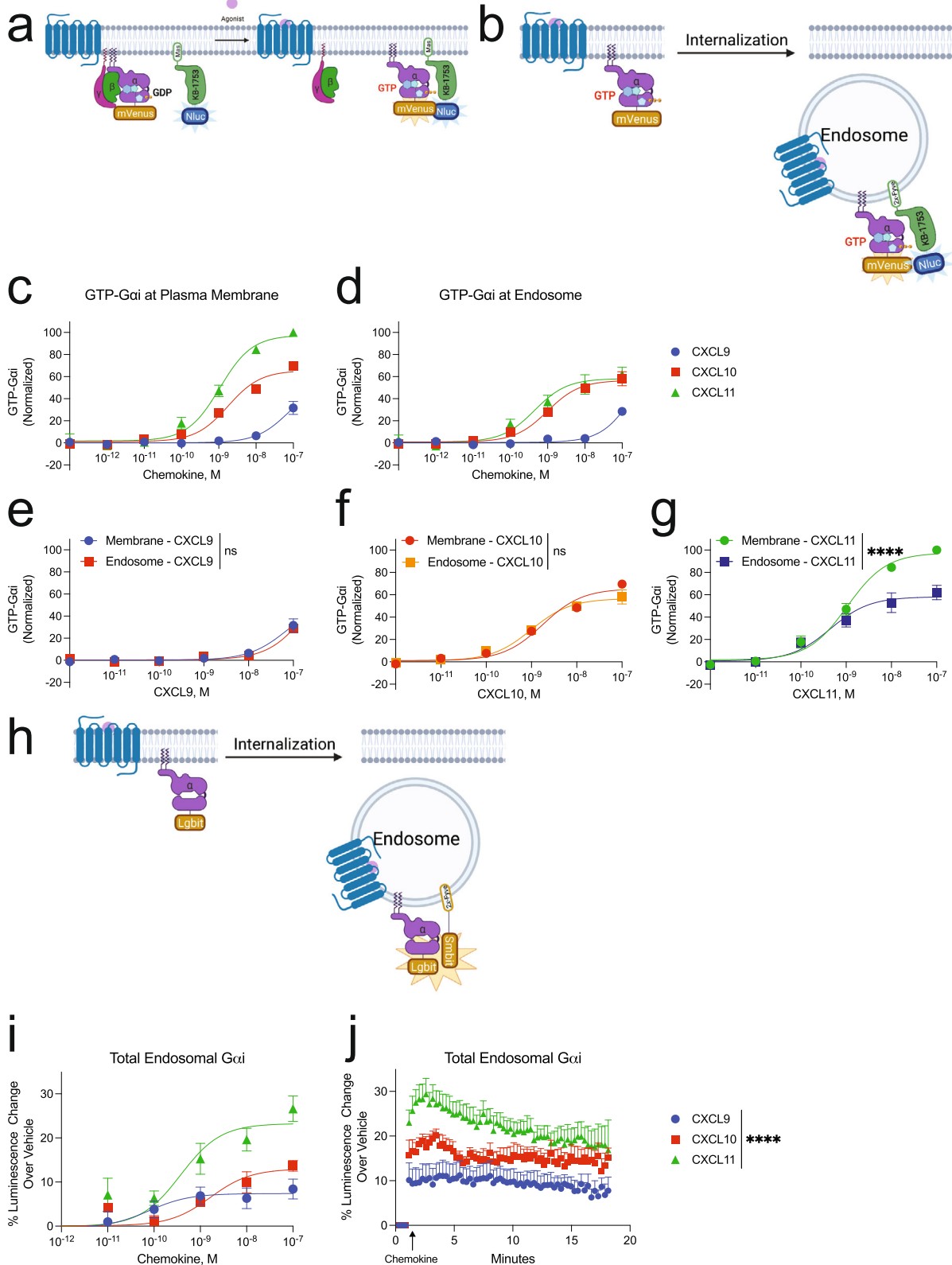

common GPCR signaling pathway[35,57]. Using Western blotting of pERK from whole cell lysates, we observed significant increases in pERK by CXCL10 and CXCL11, and relatively less activation by CXCL9 at 5 min (Fig. 5a). Upon expression of Dynamin K44A, CXCL9-induced pERK was unchanged, while CXCL10 and CXCL11 demonstrated reduced pERK levels; however, this effect was not statistically significant (Fig. 5b). Similar findings across the chemokines were observed at

30 min, and pERK levels declined back to baseline at 60 min (Supplementary Fig. 4a).

To more accurately assess ERK 1/2 activation in different subcellular locations, we generated a BRET-based biosensor of the previously developed extracellular signal-regulated kinase activity reporter (EKAR) biosensor which reports on ERK kinase activity[58] (Supplementary Fig. 4b). This biosensor can be localized to the nucleus or

**Fig. 2 | CXCR3 G protein signaling changes as the receptor traffics away from the plasma membrane to the endosome.** Schematic representation of the location-specific BRET-based GTP-Gαi sensor. Following G protein activation, the GTP-bound Gαi-mVenus will interact with the peptide KB-1753-NLuc, which selectively binds GTP-bound Gαi 1-3, to produce a BRET signal. The peptide is localized to **a** plasma membrane or the **b** endosome. Agonist dose-dependent formation of GTP-Gαi at the **c** plasma membrane as measured at 5 min or **d** endosome in HEK293 cells as measured at 15 min. Data for each ligand at the plasma membrane and endosome are presented according to ligand identity. Data for figures **c–g** are normalized to CXCL11-induced GTP-Gαi at the plasma membrane. **h** Schematic

representation of the split nanoluciferase assay detecting total endosomal Gαi irrespective of Gαi nucleotide status. **i** Agonist dose-dependent Gαi-LgBit recruitment measured between 3 and 5 min. **j** Kinetic data of Gαi-LgBit recruitment to endosomes tagged with 2xFyve-SmBit following 100 nM chemokine treatment. Data are the mean ± SEM, $n = 3$ independent plate-based experiments for **c–g** and $n = 6$ independent plate-based experiments for **I, j**, ns $P \geq 0.05$, *$P = 0.01–0.05$, **$P = 0.001–0.01$, ***$P = 0.0001$ to $0.001$, ****$P < 0.0001$ denotes statistically significant differences between $E_{max}$ of ligands or area under the curve (AUC). Extra sum of squares F test was used for **e–g** and one-way ANOVA followed by Tukey's multiple comparison's test for **j**. See also Supplementary Fig 1.

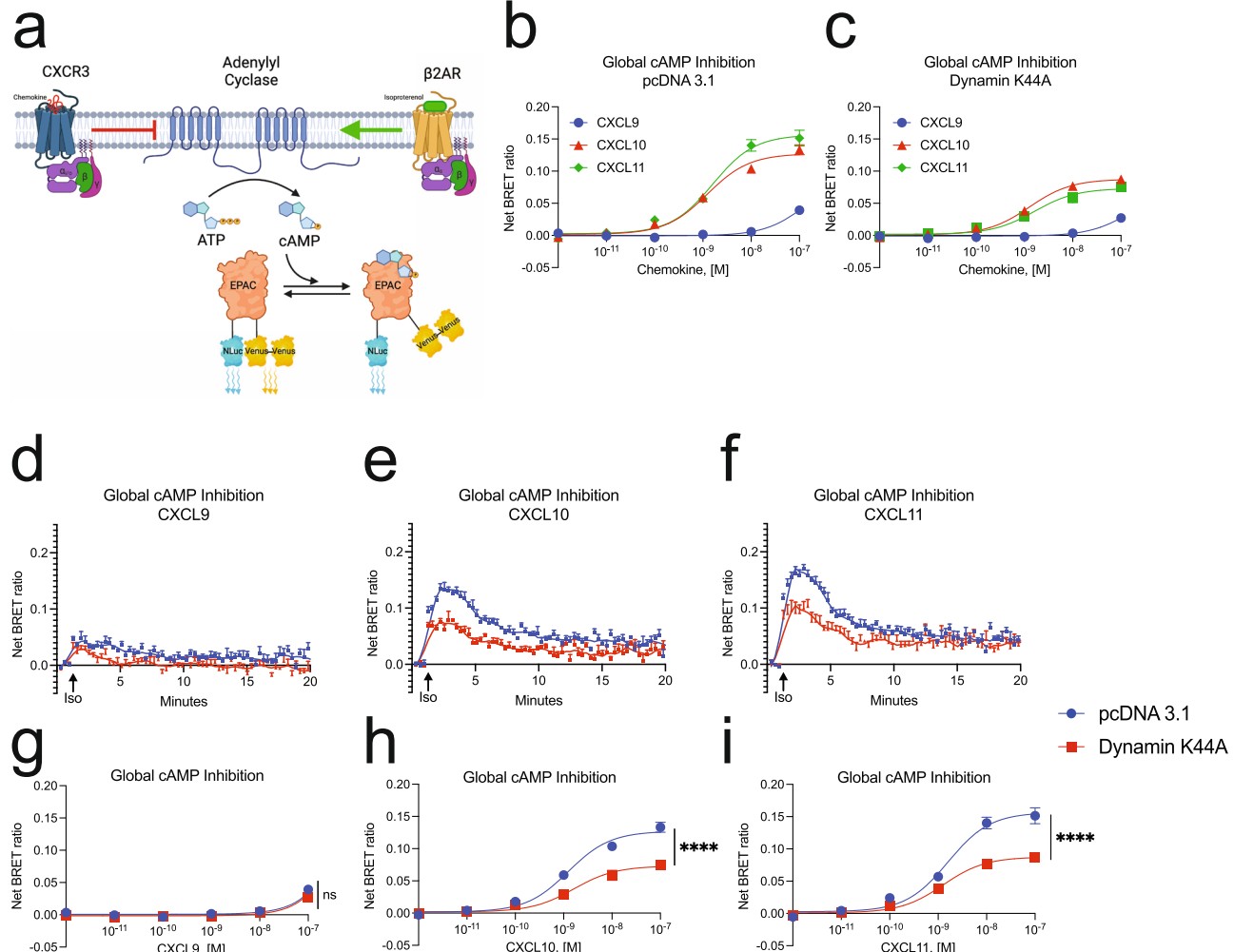

**Fig. 3 | Maximal Gαi mediated cAMP inhibition at CXCR3 is dependent on receptor endocytosis. a** Schematic representation of the cAMP sensor experimental design[40]. Agonist dose-dependent inhibition of isoproterenol-induced cAMP production by the chemokines in HEK293 cells with concurrent transfection of **b** pcDNA 3.1 or **c** Dynamin K44A inhibits internalization as measured between 3- and 5-min following addition of 1 μM isoproterenol. **d–f** Kinetic data of 100 nM treated cells and **g–i** agonist dose-dependent of cAMP inhibition levels in HEK293

cells treated with CXCL9, CXCL10, and CXCL11, respectively. Data for **g–i** are measured between 3- and 5- min following addition of isoproterenol. Data for **b–i** are the mean ± SEM, $n = 5$ independent plate-based experiments. ns $P \geq 0.05$, *$P = 0.01–0.05$, **$P = 0.001–0.01$, ***$P = 0.0001–0.001$, ****$P < 0.0001$ denotes statistically significant differences between $E_{max}$ for dose–response data of pcDNA 3.1 versus Dynamin K44A transfection conditions at each ligand. Extra sum of squares F test was used for **g–i**. See Supplementary Fig. 2 for similar data on nuclear cAMP.

cytoplasm to allow for detection of ERK activity in different subcellular compartments (Supplementary Fig. 4c, 4d). Consistent with our immunoblots, we observed biased activation of cytoplasmic ERK by the chemokines (Fig. 5c–e). Dynamin K44A partially abrogated cytoplasmic ERK activity at CXCL10 and CXCL11, as well as CXCL9, but not to a statistically significant extent (Fig. 5f). In contrast, we detected no measurable nuclear ERK activity with CXCL9 treatment, but substantial

nuclear ERK activity with CXCL10 and CXCL11. Dynamin K44A expression led to near complete abrogation of nuclear ERK activity by both CXCL10 and CXCL11 (Fig. 5g–j). These findings suggest that CXCR3 internalization is necessary for activation of nuclear ERK, while CXCR3 internalization contributes to, but is not required for, cytoplasmic ERK activation. Furthermore, while CXCL9 promotes cytoplasmic ERK activity, it does not promote measurable nuclear ERK activation.

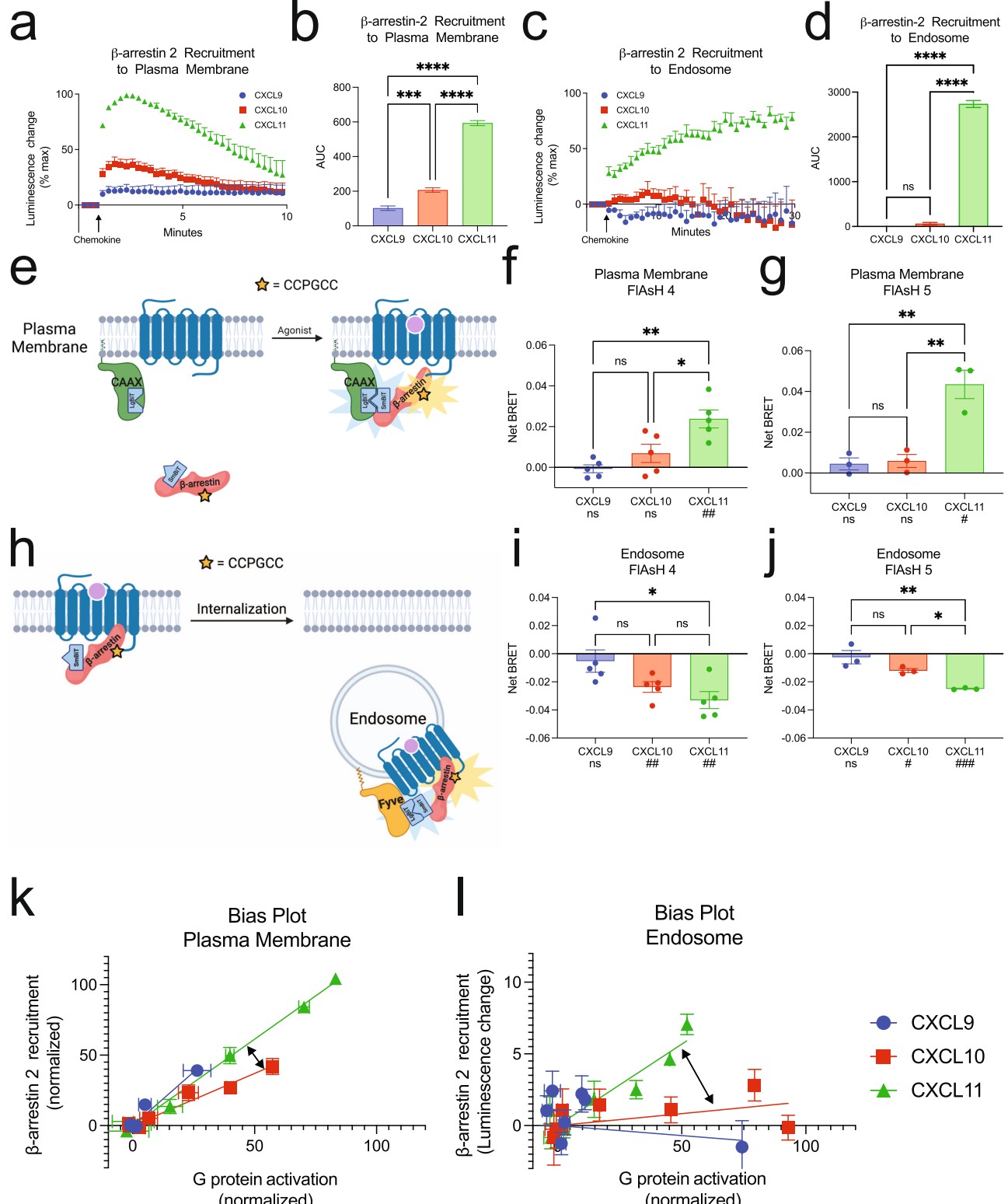

## Biased agonists are differentially dependent on internalization for transcriptional regulation

Previous studies have shown that certain transcriptional responses are dependent on sustained GPCR signaling from endosomes[9,59]. Notably, CXCL9, CXCL10, and CXCL11 have also previously been shown to differentially activate transcriptional reporters[26]. To determine the contribution of CXCR3 signaling from endosomes to the transcriptional response, we studied the chemokine-induced activation of two transcriptional reporters, the serum response element (SRE), which

responds to ternary complex factor (TCF)-dependent MAPK/ERK signaling, and serum response factor response element (SRF-RE), which is a mutant form of SRE that responds to TCF-independent signaling pathways like RhoA[60]. Consistent with previous work, CXCL11 promoted the most transcriptional activity at both reporters, followed by CXCL10 and CXCL9 in HEK293 cells (Fig. 5k, m). Overexpression of Dynamin K44A significantly decreased CXCL11-mediated transcriptional activity, but had no significant effect on CXCL9- and CXCL10-mediated transcriptional activity. Inhibition of endocytosis led to a

**Fig. 4 | CXCR3 demonstrates biased β-arrestin 2 recruitment and conformation between chemokine agonists at the plasma membrane and endosome.** Kinetic data and quantification of AUC β-arrestin-2 recruitment to the **a** and **b** plasma membrane or **c** and **d** endosome following 100 nM chemokine stimulation of CXCR3. Figures showing AUC (**b** and **d**) display the mean and SEM of the raw kinetic data presented in the preceding panels (**a** and **c**), respectively. **e** Schematic of complex FlAsH assay to detect β-arrestin 2 conformation. Cells express LgBit-CAAX and a modified SmBit-β-arrestin 2 complex FlAsH construct. Upon complex FlAsH recruitment to the plasma membrane, complementation between the LgBit and SmBit creates a functional nanoluciferase protein which can undergo BRET with the intramolecular tetracysteine motif. **f** and **g** Complex FlAsH 4 and 5 plasma membrane BRET data for CXCR3 treated with 100 nM chemokines as measured between 3- and 5-min following stimulation. **h** Schematic of complex FlAsH assay, similar to Fig. 4e, to detect β-arrestin 2 conformation at the endosome, using 2x-Fyve-LgBit. **i** and **j** Complex FlAsH 4 and 5 endosomal BRET data for CXCR3 treated with 100 nM chemokines as measured between 15- and 20- min following stimulation. **k** and

**l** Bias plots demonstrating relative G protein activation and β-arrestin 2 recruitment at the plasma membrane and endosome across the chemokines. Arrows highlight the change in best fit lines between CXCL10 and CXCL11. For β-arrestin 2 recruitment assays, data are the mean ± SEM, $n = 4$ independent plate-based experiments. ns $P \geq 0.05$, *$P = 0.01$-$0.05$, **$P = 0.001$-$0.01$, ***$P = 0.0001$ to $0.001$, ****$P < 0.0001$ denotes statistically significant differences between AUC between different chemokines. One-way ANOVA followed by Tukey's multiple comparison's test was used for **b** and **d**. For complex FlAsH assays (**f, g,** and **l, j**), data are the mean ± SEM, $n = 3$ independent plate-based experiments for experiments using complex FlAsH 5, and $n = 5$ independent plate-based experiments for experiments using complex FlAsH 4. ns $P \geq 0.05$, *$P = 0.01$-$0.05$, **$P = 0.001$-$0.01$, ***$P = 0.0001$ to $0.001$, ****$P < 0.0001$ by one-way ANOVA with Tukey's post-hoc testing conducted between ligands within each FlAsH construct. ns $P \geq 0.05$, #$P = 0.01$-$0.05$, ##$P = 0.001$-$0.01$, ##$P = 0.0001$-$0.001$, ####$P < 0.0001$ by a one-sample $t$-test is listed beneath each chemokine to determine if the Net BRET value was non-zero. See also Supplementary Fig. 3.

50% decrease in CXCL11-induced transcriptional activation, which was significantly greater than that observed at CXCL9 and CXCL10 (Fig. 5l, n). Interestingly, although CXCL10 promoted nuclear ERK activation in an endocytosis-dependent manner, inhibition of endocytosis did not impact CXCL10 activation of SRE to the same extent as CXCL11. These data suggest that CXCL10 and CXCL11 regulate transcriptional activation of this promoter element through different mechanisms, where CXCL11 demonstrates greater relative dependence on receptor internalization. Importantly, inhibition of endocytosis significantly decreased the degree of bias observed between the chemokines, demonstrating the critical role internalization plays in GPCR functional selectivity.

### Chemokine-induced transcription in CD8 + T cells reveals differential activation of inflammatory pathways

CXCR3 is primarily found in blood, bone marrow, and lymphoid tissues, specifically on Th1-type CD4 + T cells and effector CD8 + T cells[23]. To study the biased transcriptional regulation at CXCR3 in a more physiologically relevant cell type, the transcriptional response of primary, activated, CD8 + human T-cells expressing endogenous amounts of CXCR3 stimulated with the chemokines was characterized by RNA Sequencing (RNA-Seq) (Supplementary Fig. 5a, 5b). We observed significant changes in global transcriptional activation, detecting approximately 48000 transcripts, 887 of which varied by chemokine treatment (Fig. 6a). There was a high degree of replicability between biological replicates (Supplementary Fig. 5c–f). The majority of differentially expressed genes (DEGs) increased in transcript level following chemokine treatment (Supplementary Fig. 5g–k). CXCL11 demonstrated the largest number of DEGs, consistent with our data in HEK293 cells (Fig. 6b). Importantly, CXCL10 and CXCL11 demonstrate transcriptional profiles where the majority of DEGs were only found following treatment with each specific chemokine, rather than being shared across chemokines. These data contrast with that observed at CXCL9—although it promoted significant transcriptional activation, -66% of CXCL9-induced DEGs were shared with CXCL10 and or CXCL11 (Fig. 6b).

We next analyzed the DEGs by Gene Set Enrichment Analysis (GSEA) using the Molecular Signatures Database (mSigDB)[61]. GSEA identified differentially activated biological pathways and processes corresponding to predefined mSigDB gene sets. Compared to vehicle control, the chemokines induce biased activation of pathways including interleukin JAK/STAT signaling, Myc targets, and TNF-α/NF-κB, among others. Comparison of DEGs between chemokines revealed differential activation of eight gene sets between CXCL9 and CXCL10, 24 between CXCL9 and CXCL11, and 11 gene sets between CXCL10 and CXCL11 (Fig. 6c–e). Among them, several were proinflammatory including TNF-α/NF-κB, IL6/JAK/STAT3, MYC, mTORC1, and IFNγ-related pathways. CXCL11 was enriched in pathways related to the

transcription factor MYC and apoptosis, suggesting that CXCL11 plays a role in regulating T-cell growth[62]. In contrast, CXCL10 shows enrichment in cytokine-related pathways (JAK/STAT, INFγ), complement, and inflammatory responses, suggesting that CXCL10 may promote a pro-inflammatory T-cell phenotype. These findings highlight the lack of conserved transcriptional response across the chemokines, demonstrating the physiologic role of sustained signaling from endosomes in biased regulation of inflammatory pathways.

### CXCR3 internalization contributes to potentiation of inflammation in vivo

We previously showed in a murine model of allergic contact hypersensitivity (CHS) that a synthetic β-arrestin-biased CXCR3 agonist, VUF10661, potentiates inflammation through increased recruitment of CD8 + T cells in a β-arrestin 2-dependent manner[27]. To determine if this response requires sustained CXCR3 signaling from endosomes, we inhibited receptor-mediated internalization in this CHS model. Following sensitization, CHS was elicited through application of the allergen dinitrofluorobenzene (DNFB) or vehicle control to the ears of the mice with concomitant administration of VUF10661 and a pharmacologic inhibitor of Dynamin, Dyngo 4a[20,63,64]. Ear thickness was measured as a marker of inflammation (Fig. 6f). Previous work showed that VUF10661 in the absence of DNFB does not illicit an inflammatory response[27] and we observed similar findings with Dyngo 4a (Supplementary Fig. 6a). Therefore, any increase in ear thickness was primarily due to modulated DNFB-induced inflammation, and not directly from the compounds tested.

Following DNFB sensitization and treatment, mice treated with VUF10661 demonstrated a 60% increase in ear thickness over control (Fig. 6g). The maximal ear thickness observed amongst all treatment groups was observed 72–120 h following DNFB elicitation, consistent with previous reports (Supplementary Fig. 6b)[27]. This effect was decreased in mice that received concomitant administration of Dyngo 4a and VUF10661 compared to control, with only a 20% increase in ear thickness. These results are consistent with the conclusion that sustained CXCR3 signaling from endosomes is required for maximal potentiation of the inflammatory response. Together, these data demonstrate the in vivo role of subcellular GPCR signaling in modulating inflammation.

### Discussion

Our findings are synthesized in a working model of how location bias by CXCR3 chemokine agonists promote functionally selective responses with distinct effects on inflammation (Fig. 7). At the plasma membrane, the chemokines demonstrate biased engagement of G proteins and β-arrestins leading to different amounts of β-arrestin-dependent receptor-mediated endocytosis. In the endosomes, we observed relative changes in signaling across all

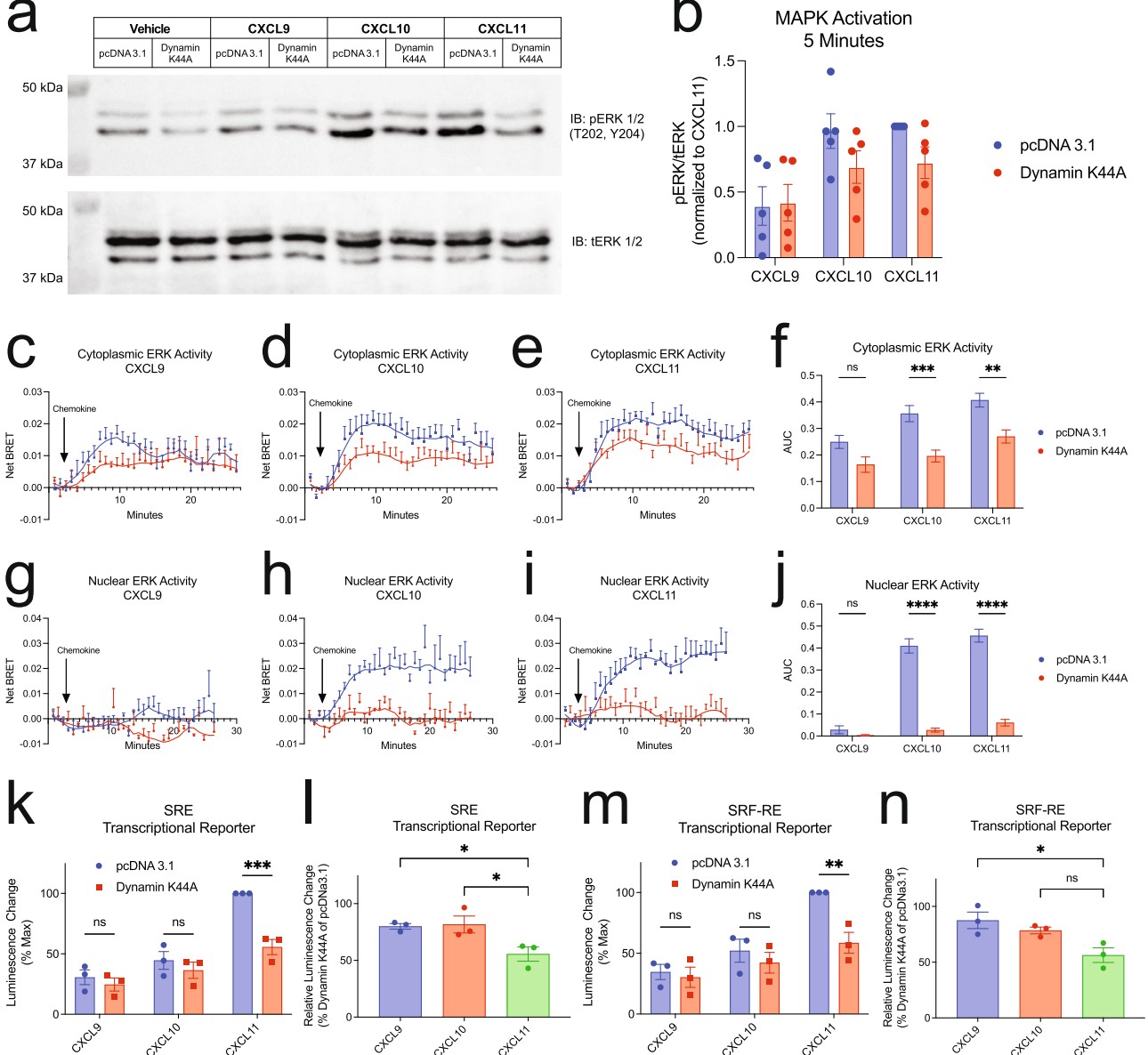

**Fig. 5 | CXCR3 internalization is required for biased cytoplasmic and nuclear ERK1/2 activation and cellular transcription. a** Representative western blot and **b** quantification of ERK1/2 phosphorylation following 5 min of stimulation with vehicle control or 100 nM chemokine with transfection of pcDNA 3.1 or Dynamin K44A. Data are the mean ± SEM, $n = 5$ independent experiments, and are normalized to CXCL11 and pcDNA 3.1. Kinetic data and quantification of AUC of ERK activity using the **c–f** cytoplasmic and **g–j** nuclear ERK BRET biosensors following 100 nM chemokine treatment with transfection of pcDNA 3.1 or Dynamin K44A. Data are the mean ± SEM, $n = 7$ independent plate-based experiments for cytoplasmic ERK sensor (**c–f**) and $n = 4$ independent plate-based experiments for nuclear ERK sensor (**g–j**). ns $P ≥ .05$, *$P = 0.01–0.05$, **$P = 0.001–0.01$, ***$P = 0.0001–0.001$, ****$P < 0.0001$ using a one-way ANOVA analysis with comparisons made between pcDNA 3.1 or Dynamin K44A within a ligand. Figures showing AUC

(**f** and **j**) are representative of the raw kinetic data shown in the preceding Fig. 5c–e and g–i, respectively. Transcriptional activity of CXCR3-expressing HEK293 cells transfected with a (**k**) serum response element (SRE) or **m** serum response factor response element (SRF-RE) luciferase reporter and either pcDNA 3.1 or Dynamin K44A following 6-h stimulation with 100 nM chemokine. Percent of **l** SRE or **n** SRF-RE signal retained when overexpressing Dynamin K44A. For luciferase reporter assays, data are the mean ± SEM, $n = 3$ independent plate-based experiments (**k–n**). ns $P ≥ 0.05$, *$P = 0.01–0.05$, **$P = 0.001–0.01$, ***$P = 0.0001–0.001$ using a two-way ANOVA analysis with comparisons made between pcDNA 3.1 or Dynamin K44A within a ligand. A one-way ANOVA with Dunnet's post-hoc testing was conducted for **l** and **n** comparing treatments to CXCL11. See also Supplementary Fig. 4. See source data for unprocessed immunoblots.

chemokines, where CXCL11 became more β-arrestin-biased while CXCL9 and CXCL10 demonstrated enhanced coupling to G proteins. CXCR3 signaling from the plasma membrane and endosome both contributed to the cytosolic activation of ERK1/2; however, only CXCL10 and CXCL11 activated nuclear ERK1/2 in a manner almost entirely dependent on signaling from endosomes. This functionally selective and location-dependent signaling converged to differentially regulate transcription in both HEK293 cells

and primary CD8 + T cells, with differential effects on genes that play important roles in inflammation. Lastly, we found that inhibiting endocytosis in a CXCR3-mediated CHS model in mice significantly decreased inflammation. Together these findings suggest a physiologically important role for location bias in CXCR3 signaling that contributes to the inflammatory response.

It was previously believed that ligand:receptor interactions in the CKR family were redundant[65]. Considerable evidence has challenged

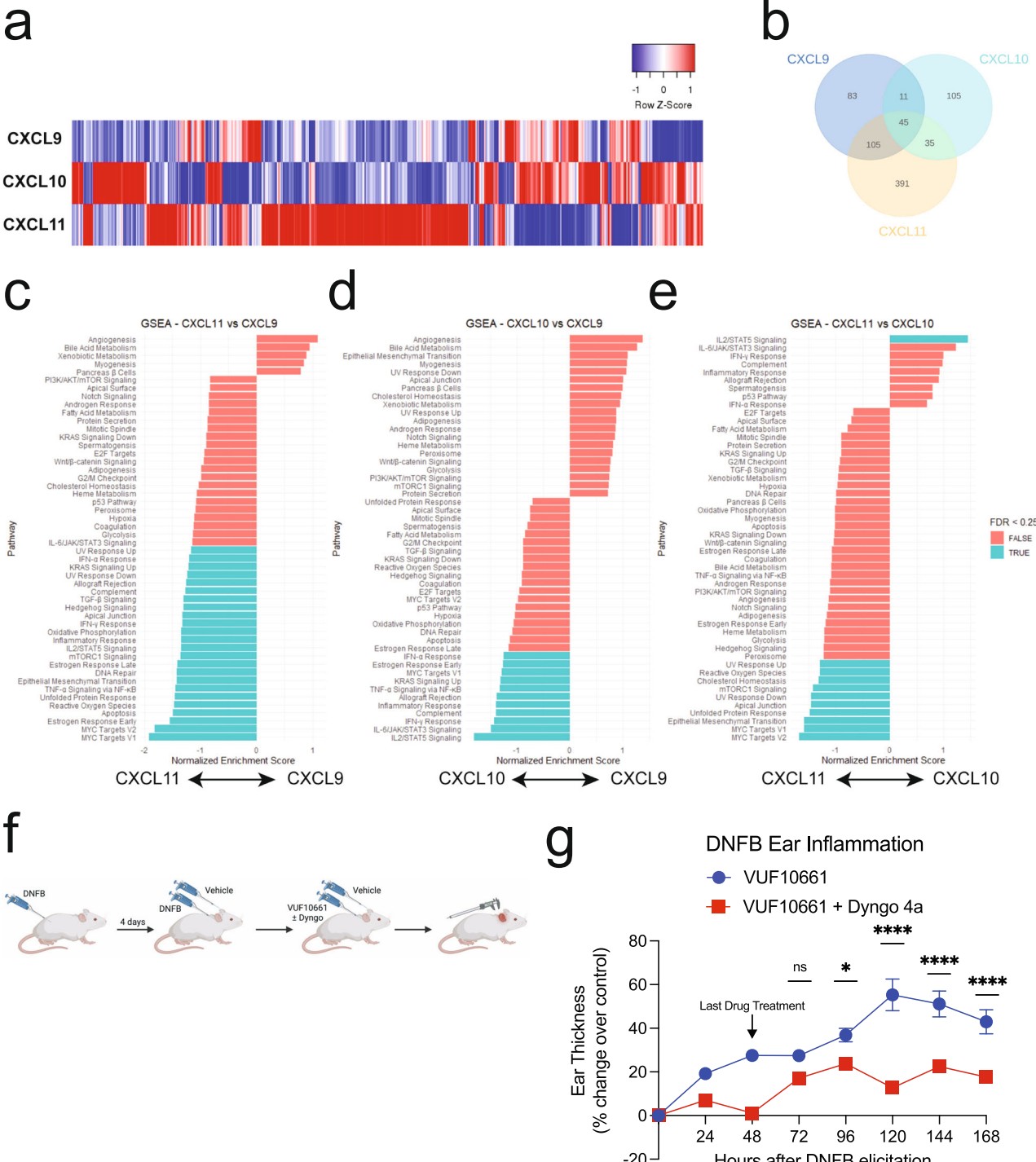

**Fig. 6 | Receptor internalization is implicated in biased transcriptional regulation in primary CD8 + T cells and required for maximum CXCR3-mediated inflammation in mice. a** Heat map of differentially expressed genes (DEGs) in primary CD8 + T-cells treated with 100 nM chemokine for 2 h. **b** Venn diagram of DEGs compared to vehicle treatment. **c**−**e** Gene set enrichment analysis of differentially regulated pathways between chemokines. Listed pathways are statistically significant at $P < 0.05$, however, select pathways are labeled as TRUE if the False Discovery Rate (FDR) is <0.25 and FALSE if the FDR is >0.25. Statistical methods for Gene Set Enrichment Analysis (GSEA) are previously described[61]. See also Supplementary Fig. 5 for additional informatics analysis of CXCR3 transcriptomics. **f** Experimental design of the dinitrofluorobenzene (DNFB)-induced contact hypersensitivity mouse model. Mice are sensitized with DNFB on their back,

followed by induction of inflammation with DNFB or vehicle control on the ears 4 days later. This is followed by treatment with VUF10661 with or without Dyngo 4a at 0, 24 h, and 48 h. **g** Ear thickness following DNFB elicitation and application of VUF10661 (50 nM) with or without Dyngo 4a (50 nM). Data are presented as the VUF10661-induced increase in ear thickness over control (DMSO or Dyngo 4a alone −see Supplementary Fig. 6 for changes in ear thickness associated with control treatments). Data are means ± SEM of mice per treatment group, $n = 8$ for vehicle/vehicle, $n = 9$ for vehicle/VUF10661, $n = 7$ for Dyngo/Vehicle, and $n = 7$ for Dyngo/VUF10661. ns $P \geq 0.05$, *$P = 0.01$–0.05, **$P = 0.001$–0.01, ***$P = 0.0001$–0.001, ****$P < 0.0001$ using a two-way ANOVA analysis with Sidak multiple comparisons testing performed at timepoints following last drug treatment.

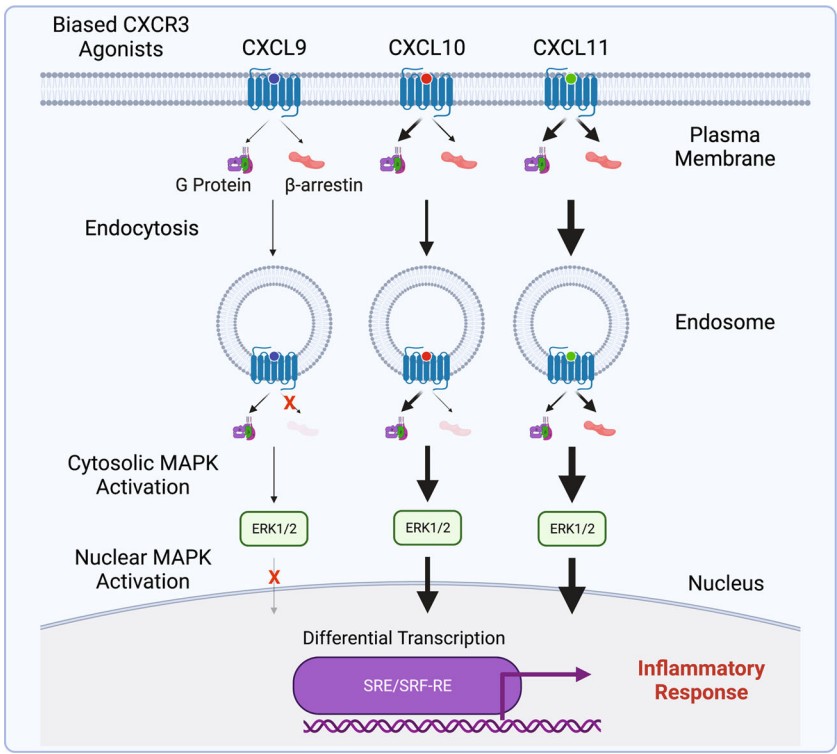

**Fig. 7 | Working model of biased subcellular signaling at CXCR3.** The three endogenous CXCR3 chemokines demonstrate unique patterns of signaling at the plasma membrane and endosome, resulting in location bias. Biased subcellular signaling was associated with differential activation of both cytosolic and nuclear ERK1/2, cellular transcription, and inflammation.

this notion and demonstrated that a significant proportion of CKR signaling is indeed specific to particular ligand:receptor combinations[47,66–68]. Here we show that the functional selectivity observed at CKRs persists beyond the plasma membrane into subcellular compartments like the endosome. Additionally, location bias is critical for some, but not all, ligands, to their functional selectivity. Given that GPCRs are known to translocate to locations like the Golgi apparatus, it is possible that trafficking to other cellular compartments may demonstrate signaling patterns different than those observed in this study[18,69,70]. Additionally, some GPCRs simultaneously exist on multiple membrane-bound structures, like the nucleus and mitochondria[18], enabling even greater signaling diversity for membrane permeable ligands.

While all of the chemokines couple CXCR3 to β-arrestin 2 at the plasma membrane, only CXCL10 and CXCL11 were able to translocate β-arrestin to endosomes, albeit to different extents. The biased chemokines also promoted unique β-arrestin conformations at the plasma membrane which persisted as the receptor trafficked to the endosome. Because β-arrestin conformation is directly related to function, it is likely these conformational differences contribute to biased receptor signaling[52]. β-arrestin can engage the GPCR core (core conformation) which is associated with G protein desensitization; however, it can also bind to the GPCR C-terminal tail (tail conformation), which is associated with receptor internalization and effector scaffolding[48,71]. There is also evidence of GPCR:G protein:β-arrestin "megaplexes" which allow for sustained G protein signaling, with simultaneous engagement of β-arrestin in the tail conformation[72,73]. We observed a relative decrease in G protein signaling in endosomes following treatment with CXCL11 but not with CXCL10. It is possible that CXCL10 promotes β-arrestin to adopt a tail conformation that drives receptor internalization without further desensitization of G protein signaling. Although CXCL11 promotes greater amounts of total endosomal β-arrestin, it is

possible that a relatively larger proportion of this β-arrestin adopts a core conformation.

Our assessments of downstream signaling demonstrate the functional diversity that can be obtained through a single GPCR using biased agonists. Biased MAPK activation observed across the CXCR3 chemokines was dependent on subcellular location. We observed significant differences in transcriptional activation that directly correspond with the ability of a ligand to activate ERK, consistent with prior studies[74]. Although overexpression of Dynamin K44A eliminated nuclear ERK activation at CXCL10 and CXCL11, we only observed a significant decrease in transcriptional activity with CXCL11 treatment. It is possible that CXCL10 and CXCL11 activate certain promoter elements through different mechanisms. This is consistent with recent reports demonstrating that some membrane-bound GPCRs can activate MAPKs via multiple mechanisms, such as translocation of Gβγ proteins to the Golgi apparatus[75].

In this manuscript, we assume that changes in signaling observed when blocking internalization were previously occurring from endosomes. While this is one interpretation of these data, it is also possible that inhibition of internalization simultaneously prolongs and/or alters signaling observed from the plasma membrane or other subcellular compartments like the late endosome, Golgi apparatus, or endoplasmic reticulum. It is difficult to quantify the absolute contribution of endosomal GPCR signaling to global GPCR signaling; however, our data demonstrate that the relative contribution of signaling from endosomes is highly dependent on the ligand used to activate the receptor, and that signaling beyond the plasma membrane is critical to the overall biased response observed in wild type cells. Further studies are needed to assess the specific signaling functions that are unique to the endosome and other subcellular structures.

Our findings highlight the critical role of ligand bias and location bias in GPCR signaling, which were further demonstrated in the diverse

transcriptional responses observed in primary CD8 + T cells and a murine model of CHS. Previous work on the Neurokinin 1 receptor (NK₁R) showed that signaling from endosomes was critical for prolonged nociception[20]. A NK₁R antagonist which trafficked with the receptor to endosomes demonstrated sustained GPCR antagonism and heightened antinociception, revealing the clinical utility of GPCR targeted therapeutics that function at multiple cellular locations[20]. We found that GPCRs can adopt multiple different signaling profiles and trafficking patterns, simply by changing the ligand used to activate the receptor. We demonstrated the potential utility of developing pharmaceutical drugs that not only activate the receptor in a biased fashion, but also target the receptor to one or multiple subcellular compartments. Given that our work was conducted at CXCR3, it is important to understand how temporospatial functional selectivity contributes to disease pathologies at other CKRs and GPCRs in order to develop more targeted, efficacious, and safer therapeutics. Because biased agonism has recently been observed at other receptor superfamilies like receptor tyrosine kinases[76], this work has important implications in harnessing the functional selectivity of chemokine receptors, GPCRs, and other transmembrane receptors at and beyond the plasma membrane.

## Quantification and statistical analysis

### Statistical analyses

Dose-response curves were fitted to a log agonist versus stimulus with three parameters (span, baseline, and EC50), with the minimum baseline corrected to zero using Prism 9.0 (GraphPad, San Diego, CA). Statistical tests were performed using a one or two-way ANOVA followed by Tukey's multiple comparison's test when comparing treatment conditions or Dunnet's multiple comparison's test when comparing treatment conditions to a control. When comparing ligands or treatment conditions in concentration–response assays, the determined $E_{max}$ of the dose-response curves fitted to a log agonist versus stimulus with three parameters was compared. When comparing time course experiments, a one-way ANOVA was conducted on the AUC. Further details of statistical analysis and replicates are included in the figure legends. Lines represent the mean, and error bars signify the SEM, unless otherwise noted.

### Bias plots

To generate bias plots, raw or normalized dose-response data for G protein activation and β-arrestin 2 are plotted for each chemokine at a specific location. We defined G protein activation as the ability of the chemokine to induce Gαi nucleotide exchange relative to the total amount of Gαi present at that location. Best fit lines were then plotted for each chemokine.

## Methods

Further information and requests for resources and reagents should be directed to and will be fulfilled by the corresponding author, Sudarshan Rajagopal (Sudarshan.rajagopal@duke.edu). All plasmids generated in this study will be distributed upon request (Supplementary Table 1).

### Bacterial strains

XL-10 Gold ultracompetent E. coli (Agilent) was used to express all constructs used in this manuscript.

### Cell lines

Human Embryonic Kidney (HEK293, β-arrestin 1/2 knockout) cells were grown in minimum essential media (MEM) supplemented with 10% fetal bovine serum (FBS) and 1% penicillin/streptomycin (P/S) at 37 °C and 5% CO2. HEK293 cells (CRL-3216) are from the American Type Culture Collection. β-arrestin ½ CRISPR/Cas9 KO HEK293 cells were generated using CRISPR/Cas9-mediated genome editing, validated

using immunoblotting, and provided by Asuka Inoue[34]. CD8 + T cells were cultured in RPMI 1640 supplemented with 10% FBS and 1% P/S at 37 °C and 5% CO₂.

### Animal studies

All animal procedures performed in this study were in agreement with the Guide for the Care and Use of Laboratory Animals of the National Institutes of Health. Animals were housed in Duke University's GSRBII and protocols for use were approved by Duke University's Institutional Animal Care and Use Committee. All animals were housed under Duke University protocol number A104-20-05. Female WT C57BL/6 (Charles River) mice were bred and maintained under specific pathogen-free conditions in accredited animal facilities at Duke University under animal protocol. Animals were housed in four mice per cage. Facilities operate between 20 and 26 °C (68–79 °F), between 30 and 70% humidity, and on a standard 12 h light/12 h dark cycle as recommend[77]. Because the ear inflammation in this CHS model causes mice to scratch and gnaw at their ears, excessive scratching can produce artificially large increases in ear thickness. To minimize this phenomenon, female mice were chosen as they tend to be less aggressive than male mice and can additionally be socially housed[78,79].

### Generation of constructs

Construct cloning was performed using conventional techniques such as restriction enzyme/ligation methods. Linkers between the fluorescent proteins or luciferases and the cDNAs for receptors, transducers, or other proteins were flexible and ranged between 2 and 18 amino acids. Dr. Kirill Martemyanov provided the EPAC plasmid which was used to clone the nuclear-specific EPAC cAMP sensors. EKAR FRET ERK1/2 biosensors previously published[58] were used to generate BRET versions of these sensors by removing the N-terminal mCerulean through restriction digest and inserting a nanoluciferase.

### Cell culture and transfection

For BRET and luminescence-based assays, HEK293 cells were transiently transfected with an optimized calcium phosphate protocol as previously described unless otherwise indicated[80]. In the calcium phosphate transfection method, cell culture media was replaced 30 min prior to transfection. Plasmid constructs were suspended in water to a final volume of 90 μL. 10 μL of 2.5 M calcium chloride was added to the plasmid constructs and mixed. 100 μL of 2× HEPES-buffered saline solution (10 mM D-Glucose, 40 mM HEPES, 10 mM potassium chloride, 270 mM sodium chloride, 1.5 mM disodium hydrogen phosphate dihydrate) was added to the solution, allowed to incubate for 2 min, and subsequently added to the cells.

For BRET biosensors for compartmentalized ERK activity and cAMP levels, transcriptional reporter assays, and confocal microscopy, cells were transiently transfected using polyethylenimine (PEI). In the PEI transfection method, cell culture media was replaced 30 min prior to transfection. Plasmid constructs were suspended in Opti-MEM (GIBCO) to a final volume of 100 μL and, in a separate tube, PEI at a concentration of 1 mg/mL was added to Opti-MEM to a final volume of 100 μL. For experiments in this manuscript, 3 μL of PEI was used per 1 μg of plasmid DNA. After 5 min, the 100 μL PEI solution was added to the 100 μL DNA solution, gently mixed, and allowed to incubate at room temperature for 10–15 min, after which the mixture was added to the cells.

### BRET and split luciferase assays

For all BRET and Split Luciferase assays, HEK293 cells seeded in six well plates were transiently transfected using the calcium phosphate method described previously unless otherwise indicated. To determine G protein nucleotide status, we took advantage of and modified a previously described two-component BRET sensor[36]. The first component of the biosensor consists of a plasma membrane targeting

domain anchor, a synthetic peptide KB-1753 that selectively and reversibly binds to GTP-bound Gαi (Gαi 1–3)[37], and a nanoluciferase BRET donor. By altering the identity of the lipid anchor, the sensor can be used to detect G protein activation at different cellular locations. Specifically, the GTP-bound Gαi sensor located at the plasma membrane (Mas-KB1753-NLuc) has a myristic attachment sequence (mas) targeting sequence (MGSSKSKTSNS)[36]. We generated a GTP-bound Gαi sensor with a 2x-Fyve targeting sequence from the hepatocyte growth factor-regulated tyrosine kinase substrate to target it to the endosome (2xFyve-KB1753-NLuc). When co-expressed with Gαi-mVenus, the sensor will bind to the active Gαi subunit following guanine nucleotide exchange of GDP for GTP and produce a BRET signal. G protein localization to endosomes irrespective of nucleotide status was detected using wild-type CXCR3, Gαi-LgBit, and 2xFyve-SmBit. The role of β-arrestin in receptor internalization was assessed using wild-type CXCR3 tagged with a C-terminal RLuc2, Myrpalm tagged mVenus or 2x-Fyve tagged mVenus, and rescue of β-arrestin 1, β-arrestin 2, both β-arrestin isoforms, or pcDNA 3.1 control. β-arrestin recruitment was assessed using wild-type CXCR3, SmBit- β-arrestin 2, and either 2xFyve-LgBit to detect β-arrestin 2 at endosomes or LgBit-CAAX to detect β-arrestin 2 at the plasma membrane. Location-specific BRET biosensors of downstream signaling (EPAC and EKAR) were transfected using PEI. The EPAC-based BRET biosensor[40] consists of an N-terminal nanoluciferase and two C-terminal Venus constructs. Following the production of cAMP by the endogenously expressed Gαs-coupled β2-adrenergic receptor (β2AR), the BRET biosensor will bind cAMP and undergo a conformational change which leads to a decrease in BRET efficiency. The EKAR biosensor consists of a target substrate that, following phosphorylation by activated pERK, binds to a phosphorylation binding domain, causing a conformational change in the biosensor and subsequent change in BRET efficiency. Twenty-four hours after transfection, cells were washed with phosphate-buffered saline, collected with trypsin, and plated onto a clear bottom, white-walled, 96 well plate at 50,000–100,000 cells/well in clear minimum essential medium supplemented with 2% FBS, 1% P/S, 10 mM HEPES, 1x GlutaMax, and 1x Antibiotic-Antimycotic (Gibco). The next day, the media were removed, and cells were incubated at room temperature with 80 μL of 3 μM coelenterazine h in Hanks' balanced salt solution (HBSS) (Gibco) supplemented with 5 mM HEPES for 5–10 min before adding ligand at the appropriate concentration. For BRET assays assessing CXCR3 internalization, HEK293 cells were stimulated with 100 nM of each chemokine and the data shown are average Net BRET ratios calculated between 25 and 30 min following stimulation. For EPAC assays, 100 nM chemokine and coelenterazine were added simultaneously and allowed to incubate for 15 min prior to the addition of 1 μM isoproterenol to promote cAMP formation. For split luciferase assays to assess Gαi-Lgbit and SmBit-β-arrestin 2 trafficking, as well as BRET EKAR and EPAC assays, three initial reads were taken prior to the addition of ligand to quantify baseline luminescence or BRET before adding ligand. Plates were read with a BioTek Synergy Neo2 plate reader set at 37 °C. All readings were performed using a kinetic protocol. BRET plates were read using a 480 nm wavelength filter and 530 nm wavelength filter. BRET ratios were calculated by dividing the 530 nm acceptor signal by the 480 nm donor signal. Net BRET ratios were calculated by subtracting the vehicle BRET ratio from the ligand stimulated BRET ratio. Split luciferase plates were read without a wavelength-specific filter. Baseline luminescence was subtracted from each read following ligand addition to calculate a change in luminescence after ligand stimulation and then normalized to vehicle treatment.

### Complex intramolecular fluorescent arsenical hairpin (FlAsH) BRET of β-arrestin 2

HEK293 cells seeded in six-well plates were transiently transfected with wild-type CXCR3, SmBit-tagged FlAsH 4 or 5, and either 2xFyve-LgBit or LgBit-CAAX using the calcium phosphate transfection protocol. In this complex FlAsH assay, CCPGCC tetracysteine sequences were inserted into a β-arrestin 2 construct following amino acids 225 in FlAsH 4 and 263 in FlAsH 5[52]. These tetracysteine motifs are capable of binding the organoarsenic compound FlAsH-EDT₂. The original FlAsH constructs have an N-terminal luciferase which, in the complex FlAsH assay, is replaced with a SmBit[52]. When the β-arrestin 2 complex FlAsH construct is recruited to one of the tagged intracellular locations, complementation occurs between the LgBit and SmBit, creating a functional nanoluciferase protein. The produced luminescent signal (~460 nm) can undergo resonance energy transfer (RET) with the intramolecular FlAsH-EDT₂, which serves as an acceptor moiety to produce a BRET signal (~530 nm). The efficiency of RET depends on the distance and conformation between the split nanoluciferase and FlAsH-EDT₂. Thus, this assay provides a readout of β-arrestin 2 conformation as measured between the N-terminus and two different locations on the β-arrestin 2 C-domain at specific subcellular locations. Twenty-four hours after transfection, cells were plated onto clear-bottomed, rat-tail collagen-coated, white-walled, Costar 96-well plates at 100,000 cells/well in minimum essential medium (Gibco) supplemented with 10% fetal bovine serum and 1% P/S. The following day, cells were washed with 50 μL of HBSS (Gibco). 100 μL of 2.5 μM FlAsH-EDT₂ in HBSS was added for arsenical labeling, and cells were incubated in the dark at 37 °C for 45 min. FlAsH-EDT₂ was aspirated, and the cells were washed with 130 μL of 250 μM 2,3 dimercaptopropanol (BAL) wash buffer. Cells were then incubated at room temperature with 80 μL of 3 μM coelenterazine h in Hanks' balanced salt solution (Gibco) supplemented with 20 mM HEPES for 5–10 min. Following a 5-min incubation in 37 °C, three prereads were taken to measure baseline BRET ratios. Chemokine was then added to 100 nM final concentration. Plates were read with a BioTek Synergy Neo2 using a 480 nm wavelength filter and 530 nm wavelength filter. Readings were performed using a kinetic protocol. BRET ratios were calculated by dividing the 530 nm signal by 480 nm signal. Net BRET values were calculated as described above by averaging six consecutive BRET values and normalizing them to vehicle control. Net BRET values of β-arrestin 2 conformation using the membrane tag LgBit-CAAX were calculated at 5 min following ligand stimulation, while Net BRET values using the endosome tag 2xFyve-LgBit were calculated at 20 min following ligand stimulation.

### Immunoblotting

Immunoblotting was performed as described previously[27]. HEK293 cells seeded in 12 well plates were transiently transfected with wild-type CXCR3 and either pcDNA 3.1 or Dynamin K44A using the calcium phosphate transfection method. 24 h after transfection, cells were serum starved in minimum essential medium supplemented with 0.01% bovine serum albumin (BSA) and 1% (P/S) for 16 h. The cells were then stimulated with 100 nM chemokine or vehicle control for 5, 30, or 60 min, washed with ice-cold PBS, and lysed in ice-cold RIPA buffer supplemented with phosphatase and protease inhibitors (Phos-STOP (Roche), cOmplete EDTA free (Sigma)). The samples were rotated at 4 °C for forty-five minutes and cleared of insoluble debris by centrifugation at 17,000 g at 4 °C for 15 min, after which the supernatant was collected. Protein was resolved on SDS-10% polyacrylamide gels, transferred to nitrocellulose membranes, and immunoblotted with the indicated primary antibody overnight at 4 °C. phospho-ERK at 1:1000 dilution (Cell Signaling Technology, #9106) and total ERK at 1:1000 dilution (Millipore Sigma, #06-182) antibodies were used to assess ERK activation. Horseradish peroxidase-conjugated anti-rabbit-IgG (Rockland #611-7302) or anti-mouse-IgG (Rockland #610-603-002) at 1:3000 dilution were used as secondary antibodies. The nitrocellulose membranes were imaged by SuperSignal enhanced chemiluminescent substrate (Thermo Fisher) using a ChemiDoc MP Imaging System (Bio-Rad). Following detection of pERK signal, nitrocellulose membranes

were stripped and reblotted for tERK at 1:1000 dilution. Relative ERK activation was calculated by dividing the intensity of pERK by tERK and comparing this ratio for a specific experimental condition to that of vehicle treatment. Images were processed using Image Lab 6.1 (BioRad).

## Transcriptional reporter assays—SRE and SRE-SF

HEK293 cells seeded in six-well plates were transiently transfected with SRE or SRF-RE reporter plasmids, wild-type CXCR3, and either pcDNA 3.1 or Dynamin K44A using the PEI transfection method. Twenty-four hours after transfection, cells were washed with PBS, collected with trypsin, and plated onto a clear bottom, white-walled, 96 well plate at 50,000–100,000 cells/well and starved overnight in serum-free minimum-essential media (Gibco) supplemented with 1% P/S. The cells were then incubated with 100 nM CXCL9, CXCL10, or CXCL11 for 6 h. The wells were aspirated and then incubated with 1.6 mM luciferin in Hanks' balanced salt solution (Gibco) supplemented with 20 mM HEPES for ten minutes. Luminescence was quantified at 480 nm using a BioTek Synergy Neo2 plate reader set at 37 °C. Transcriptional activity was quantified by calculating the fold-change in luminescence of ligand-treated cells from vehicle-treated cells. The fold-change was then normalized to maximum signal.

## Confocal microscopy

HEK293 cells were plated on rat-tail-collagen-coated 35 mm glass-bottomed dishes (MatTek Corporation, Ashland, MA) and transiently transfected using PEI with the listed constructs. Forty-eight hours following transfection, the cells were washed once with PBS and then serum starved for 1 h. The cells were subsequently treated with a control of serum-free media or the listed chemokine at 100 nM or VUF10661 at 10 μM for 45 min at 37 °C. Following stimulation, the cells were washed once with HBSS and fixed at room temperature in the dark in a 6% formaldehyde solution for 20 min. Cells were subsequently washed four times with PBS and then imaged. The cells were imaged with a Zeiss CSU-X1 spinning disk confocal microscope using the corresponding lasers for GFP (480 nm excitation), RFP (561 nm excitation), and mCerulean (433 nm excitation). Images were analyzed using ImageJ 2.3.0 (NIH, Bethesda, MD).

## RNA sequencing

Primary, negatively selected, CD8 + T cells were obtained commercially (*Precision for Medicine*, Bethesda, MD). T cells were cultured in RPMI medium 1640 containing 10% FBS, 1% P/S at 37 °C, and 5% CO$_2$. Prior to stimulation, T-cells were activated and expanded using anti-CD3 and anti-CD28 magnetic beads, and subsequently recultured without magnetic beads, as previous work has shown that this protocol increases T-cell count and surface expression of CXCR3[81]. Specifically, T cells were activated using CD3/CD28 T-cell Dynabeads (Thermo Fischer) at a 1:1 bead:cell ratio for three days and then cultured without Dynabeads for three more days in fresh media. Cells were starved for four hours in RPMI medium 1640 containing 0.01% BSA and 1% P/S and subsequently stimulated with vehicle or chemokine for 2 h. Total RNA was extracted using the RNeasy Plus RNA Extraction Kit (Qiagen). RNA sequencing was conducted by Novogene Co. (Beijing, China). Differential expression analysis between two conditions/groups (two biological replicates per condition) was performed using the DESeq2 R package (2_1.6.3). DESeq2 provides statistical routines for determining differential expression in digital gene expression data using a model based on the negative binomial distribution. The resulting P-values were adjusted using the Benjamini and Hochberg's approach for controlling the False Discovery Rate. Genes with an adjusted *P* value < 0.05 found by DESeq2 were assigned as differentially expressed. For heat maps, genes with an adjusted *p* value < 0.05 were considered as differentially expressed. For UpSet plots, genes with an adjusted *p* value < 0.05 and |log2(Foldchange)| > 0.3 are shown. For Volcano

plots, genes with an adjusted *p* value < 0.05 and |log2(Foldchange)| > 0.4 are labeled. Gene set enrichment analysis was performed to determine whether chemokine treatments generated significant differences for a priori-defined set of genes from the Molecular Signatures database version 7.5.1 (https://www.gsea-msigdb.org/gsea/index.jsp).

## Quantitative polymerase chain reaction (qPCR)

RNA isolated from peripheral blood mononuclear cells was reverse transcribed into complementary DNA (cDNA) using the iScript cDNA synthesis kit (Bio-Rad) according to the manufacturer's instructions. cDNA was analyzed using iTaq Universal SYBR Green Supermix (Bio-Rad) using the CXCR3 primers 5′ GCCATGGTCCTTGAGGTGAG 3′ and 5′ GGAGGTACAGCACGAGTCAC 3′ and 18s rRNA primers forward 5′ GTAACCCGTTGAACCCCATT 3′ and 5′ CCATCCAATCGGTAGTAGCG 3′. cDNA levels were measured using an Applied Biosystems 7300 Real-Time PCR system. PCR was performed first through polymerase activation and denaturation at 95 °C for 30 s. cDNA then underwent 40 cycles of denaturation at 95 °C for 15 s, and annealed, extension, and reading at 60 °C for 60 s. Data are expressed as fold change ($2^{-\Delta\Delta Ct}$) of each target gene compared to 18s rRNA, and then normalized to No Treatment control.

## DNFB contact hypersensitivity murine model

Seven-week-old mice were split into groups of 7–10 mice when sensitized. Animals were randomly assigned to treatment groups and investigators were blinded to pharmacologic treatments. Mice were initially sensitized by topical application of 50 μL of 0.5% DNFB (Sigma Aldrich) in a 4:1 acetone:olive oil solution on their shaved back. Four days later, they were challenged on their ears with 10 μL of 0.3% DNFB with or without Dyngo 4a (50 μM). 4, 24, and 48 h later, 10 μL of either vehicle control, VUF10661 (50 μM), Dyngo 4a (50 μM), or VUF10661 and Dyngo 4a (both at 50 μM) dissolved in a 72:18:10 acetone:olive oil:DMSO solution was applied to the ear by a blinded investigator. Ear thickness was measured at the listed time points with an engineer's micrometer (Standard Gage). To determine if Dyngo 4a had any effect on ear thickness in the absence of DNFB, we performed the above experiment in the absence of DNFB or VUF10661 and measured mouse ear thickness until 96 h after initial Dyngo 4a treatment.

## CXCR3 ligands

Recombinant Human CXCL9, CXCL10, and CXCL11 (PeproTech) were diluted according to the manufacturer's specifications, and aliquots were stored at −80 °C until needed for use. VUF10661 (Sigma-Aldrich) was reconstituted in dimethyl sulfoxide and were stored at −20 °C in a desiccator cabinet.

## Data availability

The RNA-seq data generated in this study have been deposited in the Gene Expression Omnibus database under accession code GSE192679. Molecular Signatures Database v 7.5.1 was used for Gene Set Enrichment Analysis studies. Source data are provided with this paper.

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

## Acknowledgements

We thank R.J. Lefkowitz (Duke University, USA), Nikoleta G. Tsvetanova (Duke University, USA), Laura M. Wingler (Duke University, USA), and Joshua Snyder (Duke University, USA) for guidance and thoughtful feedback throughout this work; N. Nazo for laboratory assistance. This work was supported by T32GM007171 (D.S.E.), the Duke Medical Scientist Training Program (D.S.E.), AHA 20PRE35120592 (D.S.E.), 1R01GM122798 (S.R.), K08HL114643 (S.R.), Burroughs Wellcome Career Award for Medical Scientists (S.R.), Duke University Dean's Summer Research Fellowship (N.B., C.C.H) and 5R01AR073858 (J.Y.Z.). Graphical figures in Figs. 2a, 2b, 2h, 3a, 4e, 4h, and 6f and Supplementary Figs. 4b and 5a were created with BioRender.

## Author contributions

Conceptualization, D.S.E., S.R.; Methodology, D.S.E., S.R.; Investigation, D.S.E., N.B., C.C.H., J.G., S.K., C.H., I.C., U.P., K.Z., A.W.; Writing—Original Draft, D.S.E., N.B., C.C.H., J.G., C.H., K.Z., A.W.; Writing—Reviewing and editing, D.S.E., N.B., C.C.H., J.G., S.K., J.S.S., J.Z., and S.R.; Visualization, D.S.E. and S.R.; Supervision and funding acquisition, J.Z. and S.R.

## Competing interests

The authors declare no competing interests.
