## [Peer Review File · Nature Communications]

**REVIEWER COMMENTS**

**Reviewer #1 (Remarks to the Author):**

**In this manuscript Eiger et al, describe the relevance of signaling via CXCR3 from**
**intracellular compartments following binding by the three chemokines CXCL9,**
**CXCL10 and CXCL11, which display differential biased agonism. The mechanisms**
**and outcome of biased agonistic signaling of CXCR3 are analyzed in vitro in a model**
**of HEK-293 cells. The translational relevance of the results obtained in vitro is further**
**explored in vivo in a model of mouse contact hypersensitivity (CHS) and ex vivo in**
**human CD8 T cells. Experiments in vitro are elegant and support the conclusions of**
**the manuscript. However, the correlations between the results obtained in vitro with**
**those obtained in human CD8 T cells and in the model of mouse CHS are poorly**
**presented.**

**Major concerns**

**Figure 6a should further confirm at the protein level, that the differences in the**
**transcripts correlate with the differential activation of MAPK / cAMP / ERK signaling**
**pathways.**

**Figure 6g needs to include a control group of mice treated with DNFB alone to be**
**compared with the group treated with DNFB plus VUF1066.**

**It is not clear what are the values compared in the statistical analysis of Fig. 6g. The**
**figure should specify the statistic comparisons between measurements of ear**
**thickness evaluated at 48, and 120 hours after CHS elicitation and indicate when there**
**are not significant differences between treatments.**

**The authors should explain why in the CHS of mice receiving treatment with**
**VUF10661 in the absence of Dyngo 4a the maximum increase in ear thickness is**
**observed 5 days following elicitation. In wild-type mice, administration of 0.5% DNFB**
**'per se' should have induced a potent CHS between 24 and 48h following elicitation.**

**Topical skin application of Dyngo 4a in wild-type mouse is not a receptor- or cell-**
**specific treatment. Besides of blocking internalization of CXCR3, the treatment has**
**the potential to inhibit endocytosis of other receptors relevant for type-1 biased**
**immunity and affect endocytosis in cells necessary for sensitization and elicitation of**
**the CHS response (i.e.: dendritic cells and macrophages). Because in figure 1 the**
**authors show that β -defensin 2 plays a relevant role in the internalization of CXCR3 in**
**HEK293 cells, to claim that the reduced CHS in mouse depends on the inhibition of**
**CXCR3 internalization in CD8 T cells, one possibility would be to perform the CHS**
**experiment in CD8 β -defensin 2 conditional KO mice.**

**Minor concerns:**

**Pg 7, line 109 add c after Fig 2.**

**Figure 1b: To better show differences in HEK293 cells treated with CXCL10, the “y”**
**axis can be presented in two segments increasing the length of the lower segment**
**ending in 50.**

**Reviewer #2 (Remarks to the Author):**

**GPCR signaling is central to numerous physiological and pathological processes and**
**the target of**
**many drug therapies. Therefore, there is enormous interest in understanding the**
**details of how**
**different ligands give rise to different cellular outcomes by signaling via the same**
**receptor. This**
**manuscript attempts to marry together two aspects of research in this area. First, it**
**has become**
**clear that different ligand can induce different signals upon binding to the same**
**receptors, a**
**phenomenon known as biased agonism. Second, it has become clear that GPCRs can**
**signal not only**
**from the plasma membrane but also from various cellular organelles, including**
**endosomes. This**
**manuscript now makes the case that the biased agonism exhibited by ligands of the**
**chemokine**
**receptor CXCR3 is not the same when this receptor signals from endosomes as when**
**the receptor**
**signals from plasma membrane. As usual, the devil is in the details, so I include below**
**my notes on**
**each section.**

**Section: Biased G protein activation depends on receptor location**

- • **The result that “CXCL10 and CXCL11 had nearly identical G protein activation at the**
**endosome but different amounts at the plasma membrane” is valid and clear from Fig**
**2c,d.**
- • **The statement that G protein activation decreased for both ligands is dependent on**
**how**
**100% is defined in these figures (which is not explained).**
- • **Furthermore, the comparison between the responses to the two ligands needs to be**
**normalised not only to the G protein concentration in each location (as done in Fig**
**2i,j) but**
**also to the concentration of receptor in each location, which has not been done here.**
**It**
**seems that the concentration of CXCR3 in the endosomal membrane is likely to be**
**higher for**
**CXCL11 than CXCL10. Therefore, normalisation may actually amplify the apparent**
**bias!**

**Section: CXCR3-mediated cAMP inhibition is differentially dependent on receptor**
**internalization**

- • **“Expression of Dynamin 146 K44A reduced inhibition of cAMP production following**
**stimulation with CXCL10 and CXCL11, but not CXCL9, reflecting a biased decrease in**
**G α icoupled activity”. This is not terribly convincing because it is based on a single**
**concentration**
**at which CXCL9 gives a measurable signal (Fig 3b,c)**
- • **“CXCL10 and CXCL11 both demonstrated a ~40% decrease in cAMP inhibition when**
**receptor**
**internalization was inhibited, even though the chemokines were able to promote**
**different**
**amounts of total receptor internalization (Fig. 3d-3i)”. This conclusion is based on**
**there**
**being a real difference between the maximal signals from CXCL10 and CXCL11 (Fig**
**3b). The**
**difference is actually quite small and may not be statistically significant.**

• Overall these concerns mean that the conclusion (“receptor internalization is critical
to the
biased regulation of second messengers across subcellular compartments”) is not
strongly
supported by the data.

Section: Biased ligands of CXCR3 promote differential patterns of β -arrestin 2
recruitment and
conformation at the plasma membrane and the endosome

• The key conclusion of this section is that CXCL10 and CXCL11 induce different
conformations
of β -arrestin 2 on endosomes compared to at the plasma membrane.

• The evidence for this is that endosome-localised versus membrane-localised
biosensors give
decreasing and increasing signals, respectively (in response to both CXCL10 and
CXCL11; Fig
4f,g,i,j). However, because the endosome-localised and membrane-localised
biosensors are
different, it is unclear to the reader whether they would be expected to give the same
signals in response to the same conformational changes. Does it not depend on the
position
and orientation of the reconstituted (LgBit-SmBit) nanoluciferase relative to the
tetracysteine motif? Control experiments and careful explanation are required so that
the
reader can understand the relationship between the signals observed and the
conformational changes deduced.

• If it turns out that there is good reason to believe that the different sensors should
report on
conformational changes in the same way, I think the authors need to explain how this
could
happen. After all, we are still talking about the same receptor bound to the same
ligands.

• The bias plots obviously are derived from the previous data so are affected by the
same
issues.

Section: CXCR3 signaling from endosomes differentially contributes to cytoplasmic
and nuclear ERK
activation

• The conclusion that “CXCR3 internalization is necessary for activation of nuclear
ERK, while
CXCR3 internalization contributes to, but is not required for, cytoplasmic ERK
activation” is
well-supported by the data.

• This is not particularly surprising and seems to be addressing a different question
from
biased agonism.

Sections on transcriptional regulation

• The data show that transcription (of certain reporters) stimulated by CXCL11 is
~50%
decreased when receptor internalisation is blocked, whereas transcription stimulated
by
CXCL9/10 is not significantly decreased. The conclusion that CXCL10 and CXCL11
stimulate
transcriptional activation by different mechanisms is supported by the data.

• Similarly, RNAseq data show that the different chemokines stimulate transcription of

different sets of genes.
• However, it is unclear whether the different mechanisms leading to these
transcriptional
differences are the same as the biased agonism observed/proposed above or
something
else, e.g., more classical full versus partial agonism (different signalling efficacies for
a
particular pathway).

**Section: CXCR3 internalization contributes to potentiation of inflammation in a murine**
**model of**

**contact hypersensitivity**

• **Blocking receptor internalisation reduces inflammation in a CXCR3-dependent**
**inflammation**
**model.**

• **This indicates that receptor internalisation is required for maximal inflammatory**
**effect, but**
**does not necessary support the conclusion that “sustained CXCR3 signaling from**
**endosomes**

**is required for maximal potentiation of the inflammatory response”**

**Overall, this manuscript extends observations on CXCR3 differential activation that**
**the authors have**

**reported in previous papers. There is no doubt that different endogenous (chemokine)**
**ligands give**

**rise to differences in the strength (efficacy) of signaling (for several readouts),**
**receptor**

**internalization, and downstream transcription. The observation that bias appears to**
**be different for**

**signalling from endosomes versus the plasma membrane is definitely interesting.**

**However, since**

**bias itself is a difference between differences (different ligands and different signaling**
**readouts), we**

**are now looking at a difference between differences between differences (this is not a**
**typo)!**

**Unfortunately, the experimental errors compound, a problem that dogs the field.**

**Finally, throughout this manuscript, the authors seem to assume that any change in**
**signaling upon**

**blocking internalisation can be taken as an indication that the signaling was**
**previously occurring**

**from endosomes. Is this the intended assumption? Are there not other possibilities? I**
**think this**

**needs to be discussed directly.**

**Minor Comments**

**1. Fig 1. The rescue experiment (Fig 1b) is not explained in the legend – please clarify.**

**Also,**

**please add times and concentrations.**

**2. In several figures, the times used (for concentration-response curves) and the**
**concentrations used (for time courses) are not give. Please add this information.**

**3. The transcriptional assays (including RNAseq) are done 2 hours after stimulation,**
**whereas**

**the signaling assays are on a time scale of minutes. It is very challenging to make a**
**direct**

**connection between the short time scale biased agonism and the longer time scale**
**transcription.**

Reviewer #3 (Remarks to the Author):

In this manuscript the authors have assessed the effects of three different endogenous CXCR3 ligands on regulating biased signaling from different subcellular locations. Most biased signaling studies have used synthetic agonists and thus the physiological relevance of those studies are not clear. The authors have cleverly chosen the endogenous agonists of CXCR3 receptor to investigate biased signaling. The experiments are nicely conducted, however, some of their data interpretations are over-stated and a few controls are missing. Here are the main issues that should be addressed:

- 1. The authors conclude that the amount of Gi activation by CXCL11 was decreased on endosomes compared to the plasma membrane (Figure 2d). They reach this conclusion by normalizing measured Gi activity to the total Gi present on endosomes. Since most of these activity measurements are based on normalized BRET, it is important to show compartment-targeted BRET pairs are expressed at equal levels. This is important as misleading conclusions could be drawn if the expression levels of KB-1753-NLuc on the endosomes were to be lower compared to the plasma membrane-localized KB-1753-NLuc. Controls showing similar expression levels of these constructs are necessary to accurately interpret these data.**
- 2. The authors further conclude that CXCL11 is beta-arrestin biased on the endosomes. In bias plot data shown in Figure 4I, the authors claim that while G protein activation by CXCL11 is decreased on the endosomes, CXCL11 is a better biased agonist for β -arrestin. However, they have clearly shown in Figure 4C that CXCL11 recruits more beta-arrestin to the endosomes. So why not using the same logic they have used to assess Gi protein activation for beta-arrestin activation (i.e. normalizing the ERK data in Figure 5 to total beta-arrestin levels on endosomes)? If ERK activity is considered a readout for beta-arrestin activation, using that logic, then CXCL10 is a much better beta-arrestin biased ligand compared to CXCL11, because it can barely recruit beta-arrestin to the endosomes but induces similar ERK activity.**
- 3. No experimental evidence has been provided to further support the observed decreased in FIAsH BRET signal at the endosome. In the discussion, the authors suggest that distinct GPCR/beta-arrestin interactions at each compartment (tail versus core) as an explanation. Can these differences be explained by different FIAsH sensors that have BRETs on different domains of beta-arrestin? A model with the current FIAsH data should be included to better demonstrate the negative BRET data.**

Associate Editor,

We thank you and the reviewers for your thoughtful feedback on our manuscript titled *Location bias contributes to functionally selective responses of biased CXCR3 agonists* (NCOMMS-22-03532-T). Below you will find a list of changes made to our manuscript to address the points raised by the reviewers and the location of these changes in our manuscript. We believe these changes greatly strengthen the quality of our manuscript.

Reviewer #1		
Comment	Response	Location in Manuscript
Figure 6a should further confirm at the protein level, that the differences in the transcripts correlate with the differential activation of MAPK / cAMP / ERK signaling pathways.	We observed in HEK293 cells that biased subcellular MAPK activation and transcriptional activation is dependent on differential receptor endocytosis. These data were the motivation for performing the RNAseq experiments in donor human primary CD8+ T-cells. We observed that CXCL11 treated T-cells demonstrated the largest degree of transcriptional activation in both CD8+ T-cells and HEK293 cells. Together, we used these data to state that the findings regarding biased endocytosis observed in HEK293 cells may extend into primary CD8+ T-cells. We do not make claims that these biased transcriptional profiles align with changes at the protein level.	N/A
Figure 6g needs to include a control group of mice treated with DNFB alone to be compared with the group treated with DNFB plus VUF1066.	Figure 6g is normalized to the control groups requested by reviewer 1. In our figure legend, we write “6g. Ear thickness following DNFB elicitation and application of VUF10661 (50nM) with or without Dyngo 4a (50nM). Data are presented as the VUF10661-induced increase in ear thickness over control (DMSO or Dyngo 4a alone – see Extended Fig. 5 for changes in ear thickness associated with control treatments).” Specifically, the data are presented as follows:  • (DNFB+Dyngo+VUF10661)/(DNFB+Dyngo+DMSO) • (DNFB+VUF10661)/(DNFB+DMSO) We have included the raw data for each treatment group in the Extended Data Figure 6.	Extended Data Figure 6
It is not clear what are the values compared in the statistical analysis of Fig. 6g. The figure should specify the statistic comparisons between measurements of ear thickness evaluated at 48, and 120 hours after CHS elicitation and indicate when there are not significant differences between treatments.	We have modified Figure 6g to demonstrate statistical significance at each measured point and updated the figure legend as follows: *P < .05 using a two-way ANOVA analysis with Sidak multiple comparisons testing performed at timepoints following last drug treatment.	Line 571 in Figure 6g
The authors should explain why in the CHS of mice receiving treatment with VUF10661 in the absence of Dyngo 4a the maximum increase in ear thickness is observed 5 days following elicitation. In wild-type mice, administration of 0.5% DNFB 'per se' should have induced a potent CHS between 24 and 48h following elicitation.	All treatment groups demonstrated a robust CHS response 72-120 hours following DNFB elicitation as best seen in the newly added Extended Data Figure #6. We have added the following sentence to our manuscript The maximal ear thickness observed amongst all treatment groups was observed 72-120 hours following DNFB elicitation, consistent with previous reports (Extended Data Fig. 6b)²⁷.	Line 329 in section titled “CXCR3 internalization contributes to potentiation of inflammation in a murine model of contact hypersensitivity”
Topical skin application of Dyngo 4a in wild-type mouse is not a receptor- or cell-specific	We agree that the effects of Dyngo 4a likely extend to other receptors beyond CXCR3 that undergo dynamin-dependent endocytosis. We account for this effect by normalizing our CHS	Line 318 in section titled “CXCR3 internalization

treatment. Besides of blocking internalization of CXCR3, the treatment has the potential to inhibit endocytosis of other receptors relevant for type-1 biased immunity and affect endocytosis in cells necessary for sensitization and elicitation of the CHS response (i.e.: dendritic cells and macrophages). Because in figure 1 the authors show that β-defensin 2 plays a relevant role in the internalization of CXCR3 in HEK293 cells, to claim that the reduced CHS in mouse depends on the inhibition of CXCR3 internalization in CD8 T cells, one possibility would be to perform the CHS experiment in CD8 β-defensin 2 conditional KO mice.	mouse data as detailed above. While Dyngo 4a alone did also lead to an increase in ear thickness, we normalize for this effect as explained above to reflect the effect of Dyngo 4a on CXCR3 signaling. We have performed the suggested experiment in our previous work (reference 27). Smith, J. S., et al. (2018). "Biased agonists of the chemokine receptor CXCR3 differentially control chemotaxis and inflammation." Sci Signal 11(555). In this manuscript, we demonstrate that the enhanced inflammatory response seen with VUF10661 treatment is entirely lost in mice devoid of β-arrestin 2. Additionally, we demonstrate that the VUF10661 induced potentiation leads to increased chemotaxis of murine CD8+ and CD44+ T-cells to the site of inflammation, and this response is also lost in mice devoid of β-arrestin 2. These data demonstrate that the potentiated inflammatory responses is dependent on β-arrestin 2. We describe this in the manuscript as follows: We previously showed in a murine model of allergic contact hypersensitivity (CHS) that a synthetic β-arrestin-biased CXCR3 agonist, VUF10661, potentiates inflammation through increased recruitment of CD8+ T cells in a β-arrestin 2-dependent manner²⁷.	contributes to potentiation of inflammation in a murine model of contact hypersensitivity"
Pg 7, line 109 add c after Fig 2.	Added the suggested change	Line 108
Figure 1b: To better show differences in HEK293 cells treated with CXCL10, the "y" axis can be presented in two segments increasing the length of the lower segment ending in 50.	Added the suggested change	Figure 1
Reviewer #2		
Comment	Response	Location in Manuscript
Section: Biased G protein activation depends on receptor location The result that "CXCL10 and CXCL11 had nearly identical G protein activation at the endosome but different amounts at the plasma membrane" is valid and clear from Fig 2c,d.	N/A	
The statement that G protein activation decreased for both ligands is dependent on how 100% is defined in these figures (which is not explained).	We defined 100% as the maximum G protein activation achieved by any ligand at any location. Our data demonstrate this maximum signal to be CXCL11 at the plasma membrane. In our figure legend, we write the following: Data for figures (c-g) are normalized to CXCL11-induced GTP-Gαi at the plasma membrane.	Figure 2
Furthermore, the comparison between the responses to the two ligands needs to be normalised not only to the G protein concentration in each	We performed the requested normalization as follows:  (G protein activation normalized to maximum signal)/(G protein concentration normalized to maximum signal)/(Receptor amount in endosomes normalized to maximum signal) 	Line 130 in section titled "Biased G protein activation depends on receptor

location (as done in Fig 2i,j) but also to the concentration of receptor in each location, which has not been done here. It seems that the concentration of CXCR3 in the endosomal membrane is likely to be higher for CXCL11 than CXCL10. Therefore, normalisation may actually amplify the apparent bias!	As the amount of receptor at the plasma membrane is equivalent for all of the chemokine treatments, we did not perform this calculation for the data collected at the plasma membrane. Additionally, it is difficult to quantitatively compare the absolute amount of CXCR3 present at the plasma membrane compared to the endosome using our data. As a result, we refrained from making any comparisons between these locations and only compared the chemokines in endosomes when performing this requested normalization. This data has been added to Extended Data 1 and the following text has been added: We further normalized these data by the total amount of receptor present in the endosome and similarly found that CXCL9 and CXCL10 are relatively more efficacious at promoting endosomal G Protein activation than CXCL11 (Extended Data Fig. 1b). These data demonstrate ligand and location bias in G protein activation, with different levels of G protein activation at the plasma membrane compared to the endosome depending on the agonist.	location” and Extended Data 1
Section: CXCR3-mediated cAMP inhibition is differentially dependent on receptor internalization “Expression of Dynamin 146 K44A reduced inhibition of cAMP production following stimulation with CXCL10 and CXCL11, but not CXCL9, reflecting a biased decrease in Gaicoupled activity”. This is not terribly convincing because it is based on a single concentration at which CXCL9 gives a measurable signal (Fig 3b,c)	We agree with the reviewer and have added the following sentence: However, this result may be due to the low G protein signal generated by CXCL9, and all chemokines may demonstrate reduced inhibition of cAMP production at higher and supraphysiologic levels of chemokine.	Line 149 in in section titled “CXCR3-mediated cAMP inhibition is differentially dependent on receptor internalization”
“CXCL10 and CXCL11 both demonstrated a ~40% decrease in cAMP inhibition when receptor internalization was inhibited, even though the chemokines were able to promote different amounts of total receptor internalization (Fig. 3d-3i)”. This conclusion is based on there being a real difference between the maximal signals from CXCL10 and CXCL11 (Fig 3b). The difference is actually quite small and may not be statistically significant. Overall these concerns mean that the conclusion (“receptor internalization is critical to the biased regulation of second messengers across subcellular compartments”) is not strongly supported by the data.	We agree with the reviewers comments. A limitation of this assay is that we are measuring cAMP accumulation, a second messenger of Gi activation, which is an amplified response. Thus, detecting real differences between the maximal regulation of cAMP accumulation by CXCL10 and CXCL11 is difficult. However, the data demonstrate a clear contribution of sustained CXCR3 signaling to endosomes at CXCL10 and CXCL11 in regulating second messenger accumulation. As a result, we have reworded this section and section title to reflect these limitations. New section title: CXCR3-mediated cAMP inhibition is partially dependent on receptor internalization Text added (bold): CXCL10 and CXCL11 both demonstrated a ~40% decrease in cAMP inhibition when receptor internalization was inhibited, even though the chemokines were able to promote different amounts of total receptor internalization (Fig. 3d-3i). This conclusion is dependent on there being a significant	Line 157 in in section titled “CXCR3-mediated cAMP inhibition is differentially dependent on receptor internalization”

	difference in normalized cAMP inhibition by CXCL10 and CXCL11. Given that this assay directly measures cAMP production, which is an amplified response of Gαi activity, detecting real differences between high efficacy agonists is difficult. However, these data demonstrate that both CXCL10 and CXCL11 require receptor internalization to achieve maximal inhibition of cAMP production.	
Section: Biased ligands of CXCR3 promote differential patterns of b-arrestin 2 recruitment and conformation at the plasma membrane and the endosome The key conclusion of this section is that CXCL10 and CXCL11 induce different conformations of b-arrestin 2 on endosomes compared to at the plasma membrane.	N/A	
The evidence for this is that endosome-localised versus membrane-localised biosensors give decreasing and increasing signals, respectively (in response to both CXCL10 and CXCL11; Fig 4f,g,i,j). However, because the endosome-localised and membrane-localised biosensors are different, it is unclear to the reader whether they would be expected to give the same signals in response to the same conformational changes. Does it not depend on the position and orientation of the reconstituted (LgBit-SmBit) nanoluciferase relative to the tetracysteine motif? Control experiments and careful explanation are required so that the reader can understand the relationship between the signals observed and the conformational changes deduced.	We agree with the reviewer that the ultimate conformation depends on the interaction between β -arrestin and the location marker. Given that the location markers are inherently different, this could be the source of differences observed in β -arrestin conformation at the two locations, rather than a true change in conformation. As a result, we can confidently compare the differences between ligands at one location, but comparisons across locations should be made with caution. We have modified the text as follows to reflect this limitation: While the β-arrestin 2 conformation demonstrated an increase in BRET signal at the plasma membrane, we observed a decrease in BRET signal at the endosome, which could be due to differences in β-arrestin 2 conformation at the endosome compared to the plasma membrane and/or a change in orientation between β-arrestin 2 and the different location markers.	Line 196 in “Biased ligands of CXCR3 promote differential patterns of β -arrestin 2 recruitment and conformation at the plasma membrane and the endosome”
If it turns out that there is good reason to believe that the different sensors should report on conformational changes in the same way, I think the authors need to explain how this could happen. After all, we are still talking about the same receptor bound to the same ligands.	We agree that determining if the conformations are truly different at the plasma membrane versus endosome would require many controls and is beyond the scope paper. The conclusions we can draw from this assay are largely limited to conformational differences between chemokines at the same location. While there is evidence regarding the differences between the subcellular environment at the plasma membrane as compared to the endosome (pH, lipid composition, curvature, etc.), there is not sufficient evidence in the literature to suggest which specific factor(s) would lead to a change in β -arrestin conformation.	N/A

	There is recent structural and functional evidence demonstrating that the C-edge of β-arrestin interacts with the plasma membrane, and membrane phosphoinositides can stabilize GPCR-arrestin complexes. Given that the membrane composition of the plasma membrane is significantly different from the endosome, e.g., endosomes lack phosphoinositides, this could be one potential mechanism underlying different subcellular β-arrestin conformations. However, we refrain from these discussion because the main conclusions we wish to highlight are conformational differences between chemokines at the plasma membrane and endosome. https://www.nature.com/articles/ncomms14258 https://www.nature.com/articles/s41586-020-1954-0 https://www.biorxiv.org/content/10.1101/2021.10.09.463790v2)	
The bias plots obviously are derived from the previous data so are affected by the same issues.	The reviewer is correct in that the data included in the bias plots are affected by the same issue. However, determination of bias is relative, and these concerns affect all of the chemokines. As a result, comparisons between the chemokines allows us to normalize for these concerns as we are comparing the difference across the chemokines between two locations (i.e. the difference of differences). We have added the following sentences to address this concern: Measurements in G protein activation/recruitment and β-arrestin recruitment at different locations are potentially impacted by the biosensor used to detect these events. For example, the absolute change in BRET signal of CXCL11 mediated G protein activation at the plasma membrane and endosome are different, but it is possible that this difference is due to using two different location specific biosensors, rather than amounts of activated G protein. However, we did not observe such a difference in CXCL10 mediated G protein activation at these two locations. Calculating and comparing the difference of differences (differences between chemokines and locations), removes any potential contribution that the location-specific biosensors may have on detecting cellular events (Fig. 4k-4l). In the bias plots, we highlight these difference of differences as relative changes of bias, rather than absolute changes of bias. Specifically, the slopes of the bias plots show a different rank order for the three ligands at the plasma membrane versus the endosome. This analysis, which avoids direct comparison of biosensors in different compartments, demonstrates that biosensor location-specific effects cannot account for the observed signaling biases.	Line 208 in “Biased signaling profiles of the chemokines changes as the receptor traffics to endosomes”
Section: CXCR3 signaling from endosomes differentially contributes to cytoplasmic and nuclear ERK The conclusion that “CXCR3 internalization is necessary for activation of nuclear ERK, while CXCR3 internalization contributes to, but is not required for, cytoplasmic ERK activation” is well-supported by the data. This is not particularly surprising and	We hypothesize that CXCR3 internalization is one factor that contributes to bias in patterns of cytoplasmic vs. nuclear ERK activity, with different chemokines.	N/A

seems to be addressing a different question from biased agonism.		
Sections on transcriptional regulation The data show that transcription (of certain reporters) stimulated by CXCL11 is ~50% decreased when receptor internalisation is blocked, whereas transcription stimulated by CXCL9/10 is not significantly decreased. The conclusion that CXCL10 and CXCL11 stimulate transcriptional activation by different mechanisms is supported by the data. Similarly, RNAseq data show that the different chemokines stimulate transcription of different sets of genes. However, it is unclear whether the different mechanisms leading to these transcriptional differences are the same as the biased agonism observed/proposed above or something else, e.g., more classical full versus partial agonism (different signalling efficacies for a particular pathway).	It is likely that multiple factors contribute to the observed changes in RNAseq data, which include bias as well as classic partial versus full agonism. However, our data demonstrate that not only do these chemokines demonstrate differing biased agonism, but the relative degree of biased agonism depends on the specific locations in the cell, highlighting the complexity of biased GPCR signaling. We use this RNAseq data set as supporting evidence that biased agonism is not a phenomenon limited to what we observed in non-physiological cell lines but extends to physiologically-relevant cells.	N/A
Section: CXCR3 internalization contributes to potentiation of inflammation in a murine model of contact hypersensitivity  · Blocking receptor internalisation reduces inflammation in a CXCR3-dependent inflammation model. · This indicates that receptor internalisation is required for maximal inflammatory effect, but does not necessary support the conclusion that “sustained CXCR3 signaling from endosomes is required for maximal potentiation of the inflammatory response” Overall, this manuscript extends observations on CXCR3 differential activation that the authors have reported in previous papers.	N/A	N/A

There is no doubt that different endogenous (chemokine) ligands give rise to differences in the strength (efficacy) of signaling (for several readouts), receptor internalization, and downstream transcription. The observation that bias appears to be different for signalling from endosomes versus the plasma membrane is definitely interesting.		
However, since bias itself is a difference between differences (different ligands and different signaling readouts), we are now looking at a difference between differences between differences (this is not a typo)! Unfortunately, the experimental errors compound, a problem that dogs the field. Finally, throughout this manuscript, the authors seem to assume that any change in signaling upon blocking internalisation can be taken as an indication that the signaling was previously occurring from endosomes. Is this the intended assumption? Are there not other possibilities? I think this needs to be discussed directly.	We agree that we are reporting on the difference between differences between differences; however, we believe that this approach normalizes for experimental error. Specifically, if the biosensors used in different locations in this manuscript inherently add in variability to our assays, by comparing the difference of differences (i.e. differences between ligands across locations), we can effectively normalize for this experimental variability, as we described and addressed above. Moreover, this does not change the major finding of our manuscript: that biased ligands induce different patterns of subcellular signaling (“location bias”) that are associated with different patterns of intracellular signaling and physiological responses. We specifically utilized endogenous biased ligands to (1) allow for normalization of this potential experimental limitation and (2) demonstrate that the degree of relative biased agonism depends on the specific subcellular locations being compared. We agree with the reviewers final point that we are assuming that any change in signaling upon blocking internalization is taken as an indication that the signaling was previously occurring from endosomes. We have attempted to use controls to account for this limitation. However, there are other possible explanations for our findings, and we have added the following text to our discussion: In this manuscript, we assume that changes in signaling observed when blocking internalization were previously occurring from endosomes. While this is one interpretation of these data, it is also possible that inhibition of internalization simultaneously prolongs and/or alters signaling observed from the plasma membrane or other subcellular compartments like the late endosome, Golgi apparatus, or endoplasmic reticulum. It is difficult to quantify the absolute contribution of endosomal GPCR signaling to global GPCR signaling; however, our data demonstrate that the relative contribution of signaling from endosomes is highly dependent on the ligand used to activate the receptor, and that signaling beyond the plasma membrane is critical to the overall biased response observed in wild type cells. Further studies are needed to assess the specific signaling functions that are unique to the endosome and other subcellular structures.	Line 383 in “Discussion”
Minor Comments Fig 1. The rescue experiment (Fig 1b) is not explained in the legend – please clarify. Also,	We have updated the figure legend for Fig. 1 as requested. ...(b) CXCR3 trafficking away from the plasma membrane using Myrpal-mVenus in β-arrestin 1/2 knock out cells following transfection of an empty vector (pcDNA 3.1), β-arrestin 1, β-arrestin 2, or both β-arrestin 1 and β-arrestin 2.	Figure 1 Legend

please add times and concentrations.	Data are normalized to maximum signal measured between 25- and 30-minutes following 100nM chemokine treatment and are the mean \pm SEM, n=4....	
In several figures, the times used (for concentration-response curves) and the concentrations used (for time courses) are not give. Please add this information.	We have added the times and concentrations used as requested to all figures.	All Figure Legends
The transcriptional assays (including RNAseq) are done 2 hours after stimulation, whereas the signaling assays are on a time scale of minutes. It is very challenging to make a direct connection between the short time scale biased agonism and the longer time scale transcription.	It is difficult to make a direct connection between proximal and distal signaling assays that are on drastically different timescales. However, there is a significant amount of previously published data similarly demonstrating that GPCRs can modulate transcriptional activation through G proteins, β-arrestin, and various kinases like ERK and PKA. While there are likely other factors which play a role beyond canonical G protein and β-arrestin signaling, these two effectors are highly implicated in directing downstream signaling pathways like transcription. https://www.nature.com/articles/nchembio.1665 https://doi.org/10.1242/jcs.03338 https://www.nature.com/articles/1210407	N/A
Reviewer #3		
The authors conclude that the amount of Gi activation by CXCL11 was decreased on endosomes compared to the plasma membrane (Figure 2d). They reach this conclusion by normalizing measured Gi activity to the total Gi present on endosomes. Since most of these activity measurements are based on normalized BRET, it is important to show compartment-targeted BRET pairs are expressed at equal levels. This is important as misleading conclusions could be drawn if the expression levels of KB-1753-NLuc on the endosomes were to be lower compared to the plasma membrane-localized KB-1753-NLuc. Controls showing similar expression levels of these constructs are necessary to accurately interpret these data.	To address the reviewers concern, we quantified the expression levels of the luminescence emission in these experiments. The raw luminescence values are a direct read out of expression level of the KB-1743-NLuc construct. We have added the following sentence and a supplemental figure demonstrating that the expression level of the endosomal and membrane KB-1743-NLuc constructs is not significantly different: Manuscript Text Importantly, the endosomal and membrane BRET biosensor was expressed at similar levels (Extended Data Fig 1). Figure Legend Extended Data Figure 1: Raw luminescence values of KB-1753-nLuc constructs and alternative data normalization. Related to Figure 2. (A) Raw luminescence values of the KB-1753-nLuc constructs at the endosome and the plasma membrane. (B) Alternative normalization approach for assessing G protein activation. Specifically, the amount of active G protein normalized to maximum signal was divided by the amount of total G protein normalized to maximum signal which was further divided by the amount of receptor present in endosomes normalized to maximum signal. Data are the mean \pm SEM, n = 5. * denotes statistically significant differences between paired luminescence averages between the location specific nanoluciferase construct as measured using a paired t-test.	Extended Data 1 Line 114 in “Biased G Protein activation depends on receptor location”
The authors further conclude that CXCL11 is beta-arrestin biased on the endosomes. In bias plot data shown in Figure 4I, the authors claim that while G protein activation by CXCL11	The relative contributions of G protein and β-arrestin signaling to ERK activation is incompletely understood and depends on multiple factors including receptor, agonist, cellular location, and cell/tissue type, among others. This has led to a number of papers in the literature, including those from the Kostenis and Lefkowitz groups, that argue that G proteins and β-arrestins	N/A

is decreased on the endosomes, CXCL11 is a better biased agonist for β-arrestin. However, they have clearly shown in Figure 4C that CXCL11 recruits more β-arrestin to the endosomes. So why not using the same logic they have used to assess Gi protein activation for β-arrestin activation (i.e. normalizing the ERK data in Figure 5 to total β-arrestin levels on endosomes)? If ERK activity is considered a readout for β-arrestin activation, using that logic, then CXCL10 is a much better β-arrestin biased ligand compared to CXCL11, because it can barely recruit β-arrestin to the endosomes but induces similar ERK activity.	play less and more important, respectively, roles in ERK activity. In a recently published manuscript from our group in Science Signaling (https://www.science.org/doi/10.1126/scisignal.abg5203) focused on CXCR3, we demonstrate that ERK activation increases following knock down of β-arrestin 2, depending on the parental cell type used for the experiment. Similarly, using β-arrestin 1 and 2 CRISPR-KO cells, we saw a decrease in ERK activation following rescue of β-arrestin 2 in one of the cell lines, but no effect in the other cell line. We, along with other groups, have demonstrated that both G proteins and β-arrestins can contribute to ERK signaling. https://www.science.org/doi/10.1126/scisignal.aat7650 https://www.science.org/doi/10.1126/science.aay1833 Therefore, normalizing ERK activity by endosomal β-arrestin recruitment may not be warranted as the specific relationship between ERK activation and proximal signal transducers is most likely multifactorial.	
No experimental evidence has been provided to further support the observed decreased in FIAsh BRET signal at the endosome. In the discussion, the authors suggest that distinct GPCR/β-arrestin interactions at each compartment (tail versus core) as an explanation. Can these differences be explained by different FIAsh sensors that have BRETs on different domains of β-arrestin? A model with the current FIAsh data should be included to better demonstrate the negative BRET data.	We agree that the current experimental evidence does not allow us to comment on how the sign or magnitude of the FIAsh BRET signal reflect specific changes in β-arrestin conformation outside of the known association with interdomain twist. We have modified the text to reflect this limitation: While the β-arrestin 2 conformation demonstrated an increase in BRET signal at the plasma membrane, we observed a decrease in BRET signal at the endosome, which could be due to a difference in β-arrestin 2 conformation at the endosome compared to the plasma membrane and/or a change in orientation between β-arrestin 2 and the different location markers.	Line 197 in “Biased ligands of CXCR3 promote differential patterns of β-arrestin 2 recruitment and conformation at the plasma membrane and the endosome”

**REVIEWERS' COMMENTS**

Reviewer #1 (Remarks to the Author):

The authors have addressed the concerns of this reviewers in a satisfactory way

Reviewer #2 (Remarks to the Author):

This version of the manuscript is greatly improved. The authors have adequately addressed the
previous concerns and the manuscript is also easier to read and understand.

I noticed the following very minor issues that should be addressed.

1. Line 154: Change "cAMP production" to "inhibition of cAMP production"?

2. Fig 3 legend (line 506): Change to "Dynamin K44A inhibits internalization as measured..."

3. Fig 4: Panels a and c require a colour key

Reviewer #3 (Remarks to the Author):

Although the authors have made revisions to the manuscript, my main issues that I had raised last
time remain unresolved. A summary of these issues is below:

1) This manuscript fails to provide additional mechanistic information or insight or novelty beyond
what the same lab has already reported regarding the differential effects of CXCR3 chemokines on
downstream responses.

2) The overall interpretation that the noted observations are due to biased agonism is not
supported. It is not surprising that different chemokines would promote different levels of receptor
internalization resulting in differential extents of signaling and transcriptional responses. Their
conclusion that this differential signaling is due to biased agonism at the endosome versus the
plasma membrane is based on unsupported interpretations of their biosensor data. Importantly,
their own functional data does not support their interpretations. For example, data in extended
figure 2b-d suggests that CXCL9 is the weakest ligand in inhibiting nuclear cAMP, but the authors
have concluded from data in Figure 2 that "CXCL9 and CXCL10 are relatively more efficacious at
promoting endosomal G Protein activation than CXCL11". If endosomal coupling to Gi is the reason
for the observed nuclear inhibition, then one would expect that CXCL9, which is a more efficacious
Gi activator than CXCL11, would cause a more robust inhibition instead of the weakest inhibition.
In summary, the differences in activating these transcriptional responses or potentiation of
inflammatory responses are, not surprisingly, due to the effect of full versus partial agonisms, as
full agonist (CXCL11) is inducing better receptor internalization, thus more robust transcriptional
responses.

3) One point of novelty in this paper is the different conformational change that beta-arrestin
adopts on the plasma membrane compared to endosomes. This is worth further investigation,
however the authors did not provide a convincing response or explanation for what underlies these
distinct conformations.

Associate Editor,

We thank you and the reviewers for your thoughtful feedback on our manuscript titled *Location bias contributes to functionally selective responses of biased CXCR3 agonists* (NCOMMS-22-03532-T). Below you will find a list of changes made to our manuscript to address the points raised by the reviewers and the location of these changes in our manuscript. We believe these changes greatly strengthen the quality of our manuscript.

Reviewer #1		
Comment	Response	Location in Manuscript
The authors have addressed the concerns of this reviewers in a satisfactory way	N/A	N/A
Reviewer #2		
Comment	Response	Location in Manuscript
This version of the manuscript is greatly improved. The authors have adequately addressed the previous concerns and the manuscript is also easier to read and understand. I noticed the following very minor issues that should be addressed.	N/A	N/A
1. Line 154: Change “cAMP production” to “inhibition of cAMP production”?	Changed	Line 154
2. Fig 3 legend (line 506): Change to “Dynamin K44A inhibits internalization as measured...”	Changed	Line 506
3. Fig 4: Panels a and c require a colour key	Changed	Figure 4a and 4c
Reviewer #3		
This manuscript fails to provide additional mechanistic information or insight or novelty beyond what the same lab has already reported regarding the differential effects of CXCR3 chemokines on downstream responses.	Our lab and other laboratories have previously published on the differential effects on CXCR3 chemokines on downstream responses. However, this is the first time our laboratory, or any laboratory to our knowledge, has studied the differential effects of biased CXCR3 chemokines at specific subcellular locations. We believe that we provide overwhelming evidence to suggest that signaling from subcellular locations is (1) different from that observed at the plasma membrane and (2) critical to generating biased responses both in cells and in vivo . These findings are novel and, as mentioned in the Author Checklist by Reviewer #2, “will likely encourage others to test the same idea in different systems and may eventually lead to improved approaches towards drug development”. We believe these findings are substantial contributions to the field of biased agonism, GPCR signaling, and receptor pharmacology as a whole.	N/A
The overall interpretation that the noted observations are due to biased agonism is not supported. It is not surprising that different chemokines would promote different levels of receptor internalization resulting in differential extents of signaling and transcriptional	Classical receptor theory of partial versus full agonism is unable to explain the findings observed in this manuscript. Biased agonism where a ligand can demonstrate different efficacies across multiple signaling pathways relative to a reference agonist. Ultimately, the exact nomenclature used to describe the signaling depends on how ones define biased agonism. Here and in many previously published reports (PMID: 34468132, 28290478, 21610196), we define it as the ability of a ligand to activate G proteins and β -arrestin when	Section titled “Biased signaling profiles of the chemokines changes as the receptor traffics to endosomes”

responses. Their conclusion that this differential signaling is due to biased agonism at the endosome versus the plasma membrane is based on unsupported interpretations of their biosensor data. Importantly, their own functional data does not support their interpretations.

For example, data in extended figure 2b-d suggests that CXCL9 is the weakest ligand in inhibiting nuclear cAMP, but the authors have concluded from data in Figure 2 that “CXCL9 and CXCL10 are relatively more efficacious at promoting endosomal G Protein activation than CXCL11”. If endosomal coupling to Gi is the reason for the observed nuclear inhibition, then one would expect that CXCL9, which is a more efficacious Gi activator than CXCL11, would cause a more robust inhibition instead of the weakest inhibition.

In summary, the differences in activating these transcriptional responses or potentiation of inflammatory responses are, not surprisingly, due to the effect of full versus partial agonisms, as full agonist (CXCL11) is inducing better receptor internalization, thus more robust transcriptional responses.

considering the potency and efficacy of a particular ligand relative to a reference ligand. Using bias plots, we qualitatively demonstrate that these ligands demonstrate biased signaling profiles relative to one another in a manner that is location dependent. Partial agonism does not accurately explain our data (as the Response 1 vs Response 2 curves would be superimposable for partial vs full agonists).

For example, CXCL11 is significantly more efficacious at activating G proteins at the plasma membrane than CXCL10, but the two ligands are similar in nature at the endosome. Additionally, CXCL11 is a more efficacious agonist at recruiting β -arrestin at the plasma membrane than CXCL10, but at the endosome, CXCL10 demonstrates minimal β -arrestin recruitment while CXCL11 has a robust response. Given that the ratio of relative intrinsic efficacies of G protein signaling and β -arrestin recruitment between these two ligands are different depending on the subcellular location, partial agonism alone cannot explain these data.

As for the specific example mentioned in Figure 2b-d, there is a need to understand the distinction between biased signaling and partial/full agonism. We demonstrate that CXCL9 is relatively G protein-biased in the endosome while CXCL11 is relatively beta-arrestin-biased in the endosome – this statement refers to the observation that CXCL9 activates more G protein relative to recruiting beta-arrestin while CXCL11 recruitments more beta-arrestin relative to activating G proteins. This is a description of biased signaling. However, if we observe the absolute agonism of these ligands, CXCL11 activates more G proteins than CXCL9 and recruits more beta-arrestin than CXCL9 in the endosome. Biased signaling focuses on the ratio of these two pathways at one ligand in comparison to this ratio at another ligand, while agonism alone looks at the difference between two ligands across just one signaling pathway. We see a more substantial decrease in inhibition of cAMP with CXCL11 when inhibiting endocytosis because it activates more absolute amounts of G protein in the endosome.

In our manuscript, we write “Biased agonism at GPCRs is commonly assessed in terms of the relative activation between G proteins and β -arrestins, and we summarized the above findings using bias plots (**Fig. 4k-4l**)^{55,56}. Bias plots allow for simultaneous assessment of relative activity between two assays, and the best fit lines obtained for each chemokine can assess relative bias across the ligands” (Section titled **Biased signaling profiles of the chemokines changes as the receptor traffics to endosomes**).

This sentence explains that the bias plot allows for assessment of relative activity between two assays.

We have added the following sentence for clarification:

“Bias plots do not assess the absolute degree of agonism of signaling across ligands. For example, although CXCL11 is β -arrestin-biased at the endosome and CXCL9 is G protein-biased at the endosome, CXCL11 activates more absolute amounts of G protein in the endosome than CXCL9. The relative bias between the two ligands is determined when comparing both G protein and β -arrestin signaling between the two ligands. Our analysis provides an assessment of biased signaling which

	considers the intrinsic efficacy and potency of one ligand to signal across multiple pathways in reference to another ligand.”	
One point of novelty in this paper is the different conformational change that beta-arrestin adopts on the plasma membrane compared to endosomes. This is worth further investigation, however the authors did not provide a convincing response or explanation for what underlies these distinct conformations.	We appreciate the reviewers comments on the novelty of these findings and agree that these findings warrant further investigation. At this time, we are performing additional experiments to better characterize the distinct conformations of β -arrestin in different locations.